



# On the Green's function emergence from interferometry of seismic wavefields generated in high-melt glaciers: implications for passive imaging and monitoring

Amandine Sergeant[1], Małgorzata Chmiel[1], Fabian Lindner[1], Fabian Walter[1], Philippe Roux[2], Julien Chaput[3], Florent Gimbert[2], and Aurélien Mordret[4]

[1]Laboratory of Hydraulics, Hydrology and Glaciology, ETH Zürich, Zürich, Switzerland
[2]University of Grenoble Alpes, CNRS, IRD, ISTerre, Institut des Géosciences de l'Environnement (IGE), Grenoble, France
[3]Department of Geological Sciences, University of Texas El Paso, El Paso, USA
[4]Massachusets Institute of Technology, Boston, USA

**Correspondence:** Amandine Sergeant (sergeant@vaw.baug.ethz.ch)

**Abstract.** Ambient noise seismology has revolutionized seismic characterization of the Earth's crust from local to global scales. The elastic Green's function (GF) between two receivers can be reconstructed via cross-correlation of the ambient noise seismograms. An homogenized wavefield illuminating the propagation medium in all directions is a pre-requesite for obtaining accurate GF. For seismic data recorded on glaciers, this condition imposes strong limitations on GF convergence, because of

5   minimal seismic scattering in homogeneous ice. We address this difficulty by investigating three patterns of seismic wavefields: a favourable distribution of icequakes and noise sources recorded on a dense array of 98 sensors on Glacier d'Argentière (France), a dominant noise source constituted by a moulin within a smaller seismic array on the Greenland ice-sheet, and crevasse-generated scattering at Gornergletscher (Switzerland). In Glacier d'Argentière, surface melt routing through englacial channels produces turbulent water flow creating sustained ambient seismic sources and thus favorable conditions for GF esti-

10   mates. From the velocity measurements of reconstructed Rayleigh waves, we invert bed properties and depth profiles, and map seismic anisotropy, which is likely introduced by crevassing. In Greenland, we employ an advanced pre-processing scheme which include match-field processing and eigenspectral equalization of the cross-spectra to remove the moulin source signature and reduce the effect of inhomogeneous wavefields on the GF. At Gornergletscher, cross-correlations of icequake coda waves show evidence for homogenized wavefields. Optimization of coda correlation windows further promotes the GF conver-

15   gence. This study presents new processing schemes on suitable array geometries for passive seismic imaging and monitoring of glaciers.





## 1 Introduction

Passive seismic techniques have proven efficient to better understand and monitor glacier processes on a wide range of time
and spatial scales. Improvements in portable instrumentation have allowed rapid deployments of seismic networks in remote
terrain and harsh polar conditions (Podolskiy and Walter, 2016; Aster and Winberry, 2017). Studies on seismic source processes
have revealed unprecedented details about englacial fracture propagation (e.g. Walter et al., 2009; Mikesell et al., 2012), basal
processes (e.g. Winberry et al., 2013; Röösli et al., 2016a; Lipovsky et al., 2019), glacier hydrology (Bartholomaus et al., 2015;
Gimbert et al., 2016) and iceberg calving (e.g. Walter et al., 2010; Bartholomaus et al., 2012; Sergeant et al., 2016, 2018).

The subsurface structure of ice sheets and glaciers has been characterized by analysis of seismic wave propagation in ice
bodies. As a few examples, Harland et al. (2013) and Smith et al. (2017) used records of basal seismicity to measure elastic
anisotropy in two Antarctic ice streams. Lindner et al. (2018a) identified crevasse-induced anisotropy in an Alpine glacier from
velocity anomalies by analyzing icequake seismograms at seismic arrays. Walter et al. (2015) used transient seismic signals
generated in moulins to compute frequency-dependent seismic velocities through matched-field processing and estimate the
depth of the ice-to-bedrock transition beneath a seismic network deployed on the Greenland ice-sheet (GIS).

At the same time, a new approach appeared in seismology about a decade ago which explores not only earthquakes but
also ambient noise sources generated by climate and ocean activity (Ekström, 2001; Rhie and Romanowicz, 2004; Webb,
1998; Bonnefoy-Claudet et al., 2006). Shapiro and Campillo (2004) and Shapiro et al. (2005) pointed out the possibility of
using continuous noise recordings to reconstruct propagating surface waves across a seismic array and to use them for crustal
tomography in California. Other studies followed, shaping the ambient noise background into a powerful tool to constrain the
elastic properties of the illuminated medium, making it possible to image the Earth's interior from crustal (Yang et al., 2007;
Lin et al., 2008) to local scales (e.g. Lin et al., 2013; Nakata et al., 2015) and monitor e.g., seismic fault (e.g. Brenguier et al.,
2008b; Olivier et al., 2015) and volcanic processes (Sens-Schönfelder and Wegler, 2006; Brenguier et al., 2011). Moreover,
ambient noise studies have so far led to original observations of the subsurface structure and evolution with applications to
geomechanics, hydrology and natural hazard (Larose et al., 2015, and references therein).

For the cryosphere, few studies have successfully used oceanic ambient noise records from permanent broadband stations
deployed on the rocky margins of glaciers or 10 to 500 km away to monitor the subsurface processes. Mordret et al. (2016)
and Toyokuni et al. (2018) tracked the strain evolution in the upper 5 km of the Earth's crust beneath the GIS due to seasonal
loading and unloading of the overlaying melting ice mass. More recently, Zhan (2019) detected slowing down of surface wave
velocities up to 2% in the basal till layer of the largest North American glacier (Bering Glacier, 20 km wide) during surge likely
due to the switch of the subglacial drainage from channelized to distributed.

The underlying seismic interferometry techniques used in ambient noise studies are rooted in the fact that the elastic impulse
response between two receivers, the Green's function (GF), can be reconstructed via cross-correlation of ambient noise seis-
mograms recorded at the two sites (Wapenaar, 2004; Campillo et al., 2014). Seismic interferometry consists in turning one of
the two receivers $R_1$ into a virtual source and retrieving the estimated elastic response of the medium at the second receiver $R_2$.
The retrieved GF is composed of a causal part (being the direct wavefield travelling from $R_1$ to $R_2$) and an acausal part (being





the time-reversed wavefield from $R_2$ to $R_1$). The constructed GF is thus expected to be symmetric in its causal and acausal portions (referred to as "causal-acausal symmetry"). Seismic propagation properties can then be estimated by analyzing the reconstructed wavefield which travels between $R_1$ and $R_2$.

In principle, the GF estimate is obtained in media capable to hosting of equipartitioned wavefields which provide homogeneous (i.e. isotropic) seismic illumination of the medium at the two-receiver site. An equipartioned seismic wavefield can be reached in (i) the presence of equally-distributed sources around the recording network (Wapenaar, 2004; Gouédard et al., 2008b) and/or (ii) in strong-scattering settings as scatterers act like secondary seismic sources and likely create a diffuse homogenized wavefield in all propagation directions (e.g. Hennino et al., 2001; Malcolm et al., 2004; Larose et al., 2008). The
latter condition (ii) is theoretically met in the inhomogeneous Earth's crust making ambient noise interferometry applications successful. However, the lack of seismic scattering in homogeneous ice (Podolskiy and Walter, 2016) renders challenging the reconstruction of the GF from on-ice recordings. Indeed, typical glacier seismic events lack sustained scattered coda waves (Walter et al., 2015) which normally arise from seismic wave deviations on multiscale heterogeneities in the medium (see Fig. 1a and further details in Sect. 5.1). Condition (ii) can compensate for lack of condition (i). However, microseismicity generated
on glaciers is often confined to narrow regions such as crevasse margin-zones (Roux et al., 2008; Mikesell et al., 2012) or other water-filled englacial conduits (Röösli et al., 2014; Walter et al., 2015; Preiswerk and Walter, 2018; Lindner et al., 2019). This often prevents for homogeneous seismic source distributions. Nevertheless, the abundance of local seismic sources in glaciers such as near-surface crevasse icequakes and high-frequency ambient noise from glacier melt (see Sect. 2.1) indicates a considerable potential for glacier imaging and monitoring with interferometry.

Few attempts have been conducted on glaciers to obtain GF estimates from on-ice seismic recordings. Zhan et al. (2013) first calculated ambient noise cross-correlations on the Amery ice-shelf (Antarctica) but could not compute accurate GF at frequencies below 5 Hz due to the low-velocity water layer below the floating ice-shelf, which causes resonance effects and a significantly nondiffusive inhomogeneous noise field. Preiswerk and Walter (2018) successfully retrieved accurate GF on two Alpine glaciers from the cross-correlation of high-frequency ($\geq 2$ Hz) ambient noise seismograms, generated by turbulent flow
of meltwater through the glacier drainage system. However, due to localized noise sources that also change positions over time over the course of the melting season, they could not systematically obtain accurate coherent GF.

As an alternative to continuous ambient noise, Walter et al. (2015) used crevassing icequakes recorded during a one-month seismic deployment at Gornergletscher (Switzerland). They recorded thousands of point source events which offered an idealized spatial source distribution around one pair of seismic sensors and could obtain accurate GF estimates. To overcome the
situation of a skewed illumination pattern often arising from icequake locations, Lindner et al. (2018b) used virtual reflector seismology (Wapenaar et al., 2011; Weemstra et al., 2017) on a contour of receivers enclosing the region of interest. This technique proved to be efficient to suppress spurious arrivals in the cross-correlation function which emerge in the presence of unevenly distributed sources. However, this method was applied on active sources and synthetic seismograms and its viability still needs to be addressed for passive recordings.

In this study, we provide a catalogue of methods to tackle the challenge of applying passive seismic interferometry on glaciers in the absence of significant scattering and/or even source distribution. After a review on glacier seismic sources (Sect.





2.1), we investigate the GF retrieval on three glacier settings with different patterns of seismic wavefields. In a first ideal case (case A, Glacier d'Argentière in the French Alps, Sect. 3), we take advantage of a favorable (i.e. isotropic) distribution of noise sources and icequakes recorded on a dense array. In case B (GIS, Sect. 4), a dominant persistent noise source constituted by a moulin prevents the accurate estimate of the GF across the array. We use a recently proposed scheme (Corciulo et al., 2012; Seydoux et al., 2017) that involves matched-field processing to remove the moulin signature and improve the GF estimate. In case C (Gornergletscher in Swiss Alps, Sect. 5), the limited distribution of icequakes is overcome by the use of crevasse-generated scattered coda waves to obtain homogenized diffuse wavefields before conducting cross-correlations. In order to serve as a practical scheme for future studies, the three above sections are nearly independent from each other. They focus on the processing schemes to compute or improve the GF estimate (i.e. GF convergence). We refer the reader who is not familiar with more ambient noise seismic processing to the appendix sections providing details on seismic detection methods, seismic array processing and seismic velocity measurements. Finally, in the light of our analysis, we discuss suitable array geometries and measurement types for future applications of passive seismic imaging and monitoring studies on glaciers.

## 2 Material and data

### 2.1 Glacier seismic sources

Glaciogenic seismic waves readily couple with the bulk Earth, and can be recorded by seismometers deployed at local (Podolskiy and Walter, 2016) to global ranges (Ekström et al., 2003). In this study, we focus on three classes of sources waves that are typically recorded in the ablating and melting environment of active grounded glaciers and generate a mix of high-frequency seismic waves, including surface waves which are of interest for the analysis which follows. For an exhaustive inventory of seismic signals recorded in glaciated area and associated source mechanisms, we refer to the review paper of Podolskiy and Walter (2016) and Aster and Winberry (2017).

Typically on Alpine glaciers and ablating zones, the most abundant class of recorded seismicity is related to brittle ice failure (Neave and Savage, 1970). On grounded ice, near-surface crevasse formation and expansion in response to glacial stress generate the vast majority ($\sim 90\%$, e.g. Canassy et al., 2012; Röösli et al., 2014) of small-scale impulsive seismic events with about $10^2 - 10^3$ daily recorded icequakes (Fig. 1a-b). In ablating regions, daily fluctuations in surface temperature and meltwater production modulate melt-enhanced sliding and seismicity rate (Walter et al., 2009) by introducing thermal-stress and strain rate variations (Mikesell et al., 2012; Podolskiy et al., 2019; Lombardi et al., 2019) or through hydrofracturing at intermediate depth (Roux et al., 2010; Carmichael et al., 2015). Subsurface fractures have a predominantly isotropic (volumetric) source mechanism which describes the crack opening, although a few studies report that a shear failure can also be a significant fracture component (Walter et al., 2009; Heeszel et al., 2014).

Typical near-surface icequakes have magnitudes -1 to 1 and propagate a few hundred meters before falling below the background noise level (Neave and Savage, 1970). Icequake waveforms have durations of 0.1–0.2 s and energy is concentrated in the 5–50 Hz range (Fig. 1a, d). With its maximum amplitude on the vertical component, Rayleigh waves dominate the seismogram. In contrast, the prior P-wave arrival is substantially weaker and for distant events often below noise level. Rayleigh

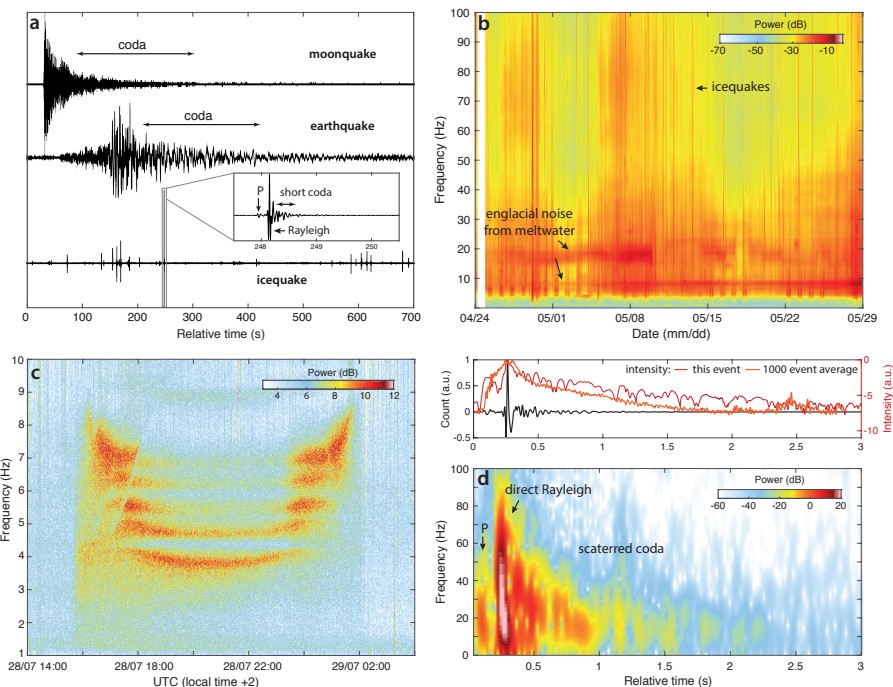

**Figure 1.** (a) Seismograms of a moonquake, regional earthquake and typical Alpine glacier seismicity. Moonquake seismogram was recorded during the 1969-1977 Apollo passive seismic experiment (Nunn, 2017). Zoom on icequake waveform shows the lack of sustained coda in homogeneous ice when compared to other signals propagating in rocky grounds. (b) Spectrogram of one month of continuous recording at Glacier d'Argentière (French Alps) showing abundance of icequakes (5-100 Hz) and englacial noise (2-30 Hz) produced by turbulent meltwater flow. (c) Spectrogram for a 10 h-long hydraulic tremor produced by the water moulin activity within the Greenland ice-sheet network (Fig.2b). (d) Spectrogram for one icequake recorded at Gornergletscher (Swiss Alps). On top of the icequake waveform, the signal intensity (see main text, Sect. 5.1) is shown for this event in red, and averaged over 1000 events in orange (right y-axis, note the logarithmic scale). Following direct Rayleigh waves, the intensity decay is linear in the late coda (at times greater than 0.5 s) which indicates scattering regime of coda waves.

waves propagate along the surface and are not excited by a source at depth exceeding one wavelength (Deichmann et al., 2000). In addition, the crevasse zone is mostly confined to the surface ($\leq$ 30 m) since ice-overburden pressure inhibits tensile fracturing at greater depths (Van der Veen, 1998). That is why such icequakes are usually considered to originate at shallow depth (Walter, 2009; Roux et al., 2010; Mikesell et al., 2012). The brief duration and weak seismic coda after the Rayleigh wave arrival (compared to earthquake coda travelling in rocky ground, Fig. 1a, see also further details in Sect. 5.1) are the

result of limited englacial scattering. This typically allows seismologists to approximate the glacier's seismic velocity model by a homogeneous ice layer on top of a rock half-space when locating events or modelling seismic waveforms (e.g. Walter et al., 2008, 2015).





From spring to the end of summer, another seismic source superimposes on icequake records and takes its origin in fluvial processes: ice melting and glacier runoff create turbulent water flow at the ice surface that interacts with englacial and sub-

glacial channels and linked conduits. The surface or subsurface gravity-driven transport of meltwater creates transient forces on the bulk Earth and surrounding ice that generate seismic waves (e.g. Schmandt et al., 2013; Gimbert et al., 2014). Analytic expressions for expected relative change of seismic power spectral density have been derived for turbulent flow in englacial channels, and allow to monitor subglacial discharge by analyzing the evolution of the channel radius and water pressure gradient from recorded seismic power over the melting season (Gimbert et al., 2016). Water flow generates a mix of seismic

waves including Rayleigh surface waves (Lindner et al., 2019) that constitute a large component of the ambient noise recorded in melting glaciers. It is recorded continuously at frequencies 1-20 Hz as shown in the spectrogram of seismic recording at Glacier d'Argentière (Fig. 1b). Seismic noise power shows diurnal variations of ambient seismicity correlated with higher discharge during daytime and reduced water pressure at night (Preiswerk and Walter, 2018).

Englacial and subglacial conduits can also generate accoustic (Gräff et al., 2019) and seismic wave resonances (Röösli et al.,

2014) as hydraulic seismic tremors generated by glacier discharge (Röösli et al., 2016b). For example, moulin tremors are produced by surface water routed through channels and moulins to the glacier base and often dominate the ambient glacier noise during peak melt hours (Röösli et al., 2014). Spectrograms of moulin tremors exhibit energy primarily between 2 and 10 Hz as recorded on the GIS (Fig. 1b). Frequency bands of either elevated or suppressed seismic energy reflect the geometry of the englacial conduit as they act as a resonanting semi-open pipe, modulated by the moulin water level (Röösli et al., 2016b).

Finally, in Alpine environments, seismic signatures of anthropogenic activity generally overlap with glacier ambient noise at frequencies > 1 Hz. Whereas anthropogenic monochromatic sources can usually be distinguished by their temporal pattern (Preiswerk and Walter, 2018), separation of all active sources recorded on glacier seismograms can prove difficult. Nevertheless, locating the source regions through matched-field processing (Corciulo et al., 2012; Chmiel et al., 2015) can help to identify the noise source processes in glaciated environments.

**2.2 Study sites and seismic experiments**

We use vertical component data of ground motion recorded on three seasonally-deployed networks in the ablation zones of two temperate Alpine glaciers and of the GIS. Each of the acquired dataset presents different patterns of seismic wavefields corresponding to the three configurations investigated for GF retrieval and defined in the introduction (cases A-C). All networks recorded varying amounts of near-surface icequakes (blue dots in Fig. 2a-c). Different processing schemes were used to

constitute the icequake catalogues and are detailed in Appendix A.

**2.2.1 Glacier d'Argentière array**

The Argentière seismic array (Fig. 2a) was deployed in late April 2018 and was recording for five weeks. It consists of 98 three-component surface sensors regularly spaced on a grid with a 350 m×480 m aperture and a station-to-station spacing of ∼ 40 m for the along-flow profiles and ∼ 50 m for the across-flow profiles. This large N-array experiment used the technology

of nodes (Fairfield Nodal Z-Land 3C) that combine a geophone, digitizer, battery, data storage, and GPS in a single box (Hand,

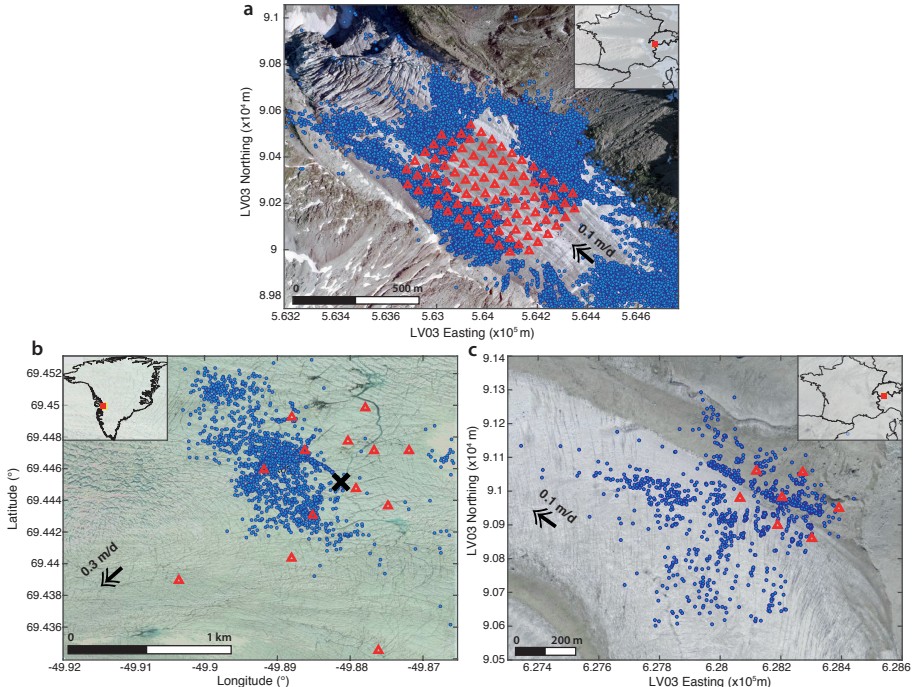

**Figure 2.** Icequake locations (blue dots) and seismic stations (red triangles) superimposed on aerial photographs of (a) Argentière (photo source: IGN France), (b) Greenland ice-sheet (photo source: © Google, Mixar Technologies) and (c) Gornergletscher (Photo source: Federal Office of Topography © swisstopo). Black arrows indicate ice flow direction. Black cross in (b) indicates the presence of a moulin within the array.

2014) and allowed a rapid deployment of the array within a few hours. Z-Land nodal sensors have natural frequency of 5 Hz and recorded continuously at a sampling rate of 500 Hz. At the array site, the ice thickness varies between 80 and 260 m (Hantz, 1981). At the time of recording, the snow layer was approximately 4-6 m thick. Ice flows at an approximate rate of 0.1 m/d and seismic sensors accumulated about 4 m of downstream displacement at the end of the experiment. Because of the snow melt,

the sensors were placed about 30 cm into the snow and leveled and reoriented twice during the deployment period. Besides seismic sensors, four on-ice GPS were deployed and a Digital Elevation Model (DEM) with a 20 m-spatial resolution. The glacier bed DEM is obtained using 14 GPR tracks conducted over the seismic array, and the glacier surface DEM is obtained from drone survey.

### 2.2.2 Greenland ice-sheet array

The GIS network (Fig. 2b) was deployed 30 km North of the calving front of the Jakobshavn Isbræfrom 2011 July 2 to August 17. The details of the study site and the seismic network can be found in Röösli et al. (2014); Ryser et al. (2014) and Andrews et al. (2014). We use seismic recordings from 13 stations: 12 seismometers (1 Hz Lennartz) installed on the



surface or shallow boreholes (2-3 m deep), and one surface broadband seismometer (Trillium Compact 120 s corner period). Seismometers recorded continuously with a sampling frequency of 500 Hz. The array has a 1.8 km aperture. It is located

around a prominent moulin with an average intake of 2.5 m$^3$/s of meltwater. At the study site, the ice is approximately 600 m-thick and flows at ∼ 0.3 m/d (Röösli et al., 2016a).

### 2.2.3  Gornergletscher array

The Gornergletscher network (Fig. 2c) operated between 2007 May 28 and July 22. It consists of 7 seismometers (six 8 Hz Geospace 11D and one 28 Hz Geospace 20D) installed in shallow boreholes (2-3 m deep). They recorded continuously with a

sampling frequency of 1000 Hz. The array has a 320 m aperture. At the study site, the ice is approximately 160 m-thick and flows at ∼ 0.1 m/d (Walter, 2009).

## 3  Passive interferometry at the Glacier d'Argentière dense array

We use a standardized processing scheme for computing cross-correlation functions at the dense array of Glacier d'Argentière. We cross-correlate seismogram time windows which encompass the seismic waves generated by discrete events in our icequake

catalogue, or cross-correlate continuous seismograms that encompass both ambient noise and icequake signals as traditionally done in ambient noise studies. Prior to any calculation, vertical components of seismic recordings are corrected for instrumental response and converted to ground velocity. Seismograms are then spectrally whitened between 1 and 50 Hz because of instrumental sensitivity at lower frequency.

For icequake cross-correlation (ICC), we follow the method of Gouédard et al. (2008b) and Walter et al. (2015). The cor-

relation time window length $T$ is adjusted to the nature of the seismic source and the array aperture to be able to encompass the icequake Rayleigh wave and to reconstruct propagating waves across the array. Here we use $T = 0.5$ s given the short icequake duration and the maximum station separation of 690 m. To avoid near-field source effects and to account for near-planar wave fronts, we only select events that lie outside a circle centered at the midpoint between the two considered stations and with a radius equal to the inter-station distance (Fig. B4a). For individual ICC (i.e. ICC computed on single events), the plane

wave approximation implies a sinusoidal dependence of surface wave arrival times with respect to event azimuth (Fig. B4b). In addition, the surface wave arrival time depends on the propagation velocity and the sensor spacing of the cross-correlated pairs. Final ICC are then obtained by stacking all individual correlation functions. To avoid any spurious arrivals in the final cross-correlation stack, Gouédard et al. (2008b) recommend to consider only sources aligned with the receiver direction (i.e. in the "endfire lobes" of the station pair), which contribute to the stationary phases of the GF. The aperture of the endfire lobes

depends on the considered seismic wavelength (Roux et al., 2004). As we here face a situation of a homogenous source distribution which covers all azimuthal directions, the ICC stack over all sources or over sources in the endfire lobes yields similar GF estimates because of canceling contribution from outside endfire loops (Walter et al., 2015).

For noise cross-correlation (NCC), we use a similar protocol as the one of Preiswerk and Walter (2018). To reduce the effects of teleseismic events or strongest icequakes, we disregard the seismic amplitudes completely and consider only one-bit

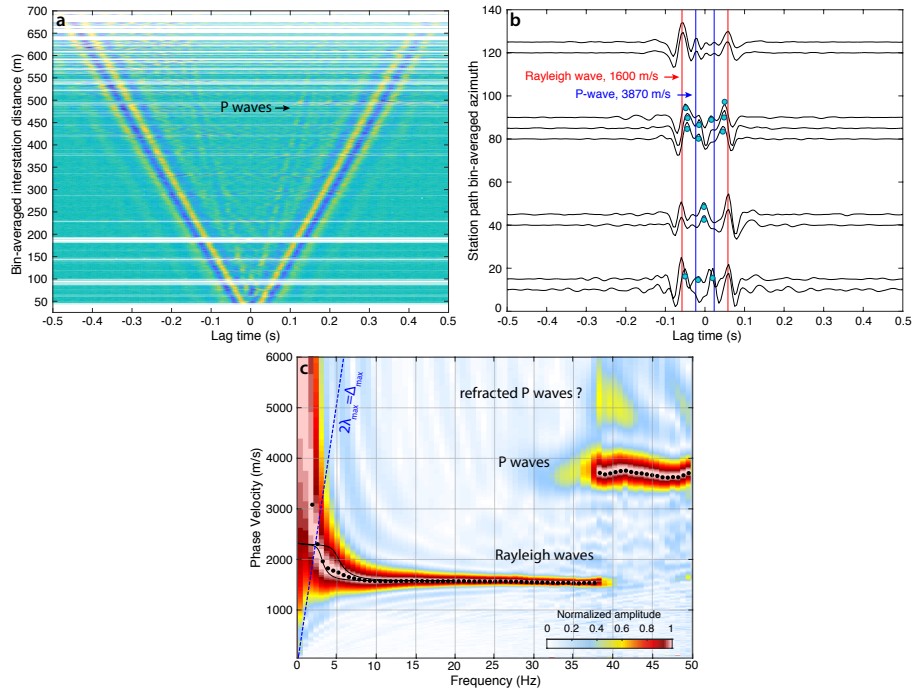

**Figure 3.** (a) Noise cross-correlations (NCC) sorted by increasing inter-station distances at Glacier d'Argentière. For the representation, correlation functions are averaged in 1 m distance bins and bandpass filtered between 10-50 Hz to highlight the presence of high-frequency P-waves. (b) Azimuthal dependence of GF estimates for pairs of stations 100 m apart. NCC converge to the GF at station paths roughly aligned with the glacier flow (azimuth $\sim 120^o$) indicating strongest noise sources located downstream and upstream of the array. For other station paths, we observe spurious arrivals (indicated by green dots) before the expected arrival times for Rayleigh (red bars) and P waves (blue bars) which primarily arise from the non-alignment of noise sources in the stationary phase zones of the stations (see main text). (c) Frequency-velocity diagram obtained from f-k analysis of NCC in (a). The dispersion curve of phase velocity for Rayleigh waves and P-waves are plotted in black dots. Dashed blue lines show the array response, given the maximum wavelength and sensor spacing $\lambda_{\max} = \Delta_{\max}/2$. Black lines are theoretical dispersion curves for fundamental mode Rayleigh wave velocity computed for ice thickness ranging from 150 m to 250 m with Geopsy software. We used the elastic parameters for the ice and bedrock as given in (Preiswerk and Walter, 2018, Sect. 6.1). Same figure for icequake cross-correlations is available in Appendix (Fig. B2).

normalized seismograms (Bensen et al., 2007). By doing so, we attribute a similar weight to ambient noise and icequake source contributions to the GF. The traces are cross-correlated in nonoverlapping 30-min long windows. Resulting NCCs are stacked daily and then averaged over the whole recording period of five weeks. We finally obtain a set of 4371 NCC that correspond to the GF estimates for all combinations of sensor pairs.



### 3.1 Green's function estimates

Figure 3a shows the stacked section of NCC averaged in 1 m-binned intervals and sorted by increasing inter-station distances. Coherent Rayleigh waves with propagation velocity of 1600 m/s are well reconstructed across the array. We also observe emergence of weak but faster waves identified at higher frequencies as P-waves traveling in the ice.

Slight disparities in amplitudes of the causal and anticausal parts of the correlation functions (positive versus negative times) are related to the source distribution. The symmetry of NCC stacked in close distance intervals indicates that noise sources are

homogeneously distributed around the network, and higher acausal amplitudes observed at larger distances are evidence for strongest sources located downstream of the array, according to our cross-correlation definition. Strongest sources downstream could be related to higher discharge near the glacier tongue (Gimbert et al., 2016) or also to the abundance of icequakes North of the array (Fig. 2a) which could influence NCC amplitudes as well. Looking closer at NCC for individual receiver pairs, we sometimes observe spurious arrivals (marked as green dots in Fig. 3b) before the expected Rayleigh wave arrival time.

Accurate GF cannot be reconstructed for a majority of pairs with directions not aligned with the glacier flow (azimuth $\sim 120^o$), indicating that dominant noise sources are mainly located downstream and upstream of the array.

The stacked section of ICC yields similar results to those of the NCC (Fig. B2). Due to the presence of icequake sources in every azimuthal direction, the ICC yield correct GF estimates at most station pairs, which supports that NCC are more sensitive to distributed noise sources rather than icequake sources. Icequake contributions certainly enable to widen the spectral content

of the NCC to frequency higher than 20 Hz, as the most energetic ambient noise is recorded in the 1-20 Hz frequency band (Fig. 1a).

Seismic phases can be identified from dispersion curves. Dispersion curves can be obtained from array processing involving frequency-wavenumber (f-k) analysis of the correlation functions obtained on a line of receivers (Appendix B1). The velocity-frequency diagram obtained from the NCC section is presented in Fig. 3c. Dispersion curves (black dots) are extracted from

the maximum intensity of the diagram at each frequency. As identified above, the correlation functions reconstruct well P-waves traveling in the ice with an average velocity $V_p = 3870$ m/s. We also observe weak intensity but fast seismic phases at frequencies above 35 Hz, which could correspond to refracted P-waves traveling along the basal interface with velocity around 5000 m/s.

Surface waves are dispersive meaning that their velocity is frequency-dependent with higher frequencies being sensitive to

surface layers and conversely, lower frequencies being sensitive to basal layers. Theoretical dispersion curves for Rayleigh wave fundamental mode are indicated in black solid lines in Fig. 3b. They correspond to a two-layer model with the top ice layer of thickness $H = 150$ m and $H = 250$ m over a semi-half space representing the bedrock. The dashed blue line indicates the array resolution capability that corresponds to the maximum wavelength limit $\lambda_{\mathrm{max}} = \Delta_{\mathrm{max}}/2$ (Wathelet et al., 2008), with $\Delta_{\mathrm{max}}$ being the maximum sensor spacing. Reconstruction of Rayleigh waves and resolution of their phase velocities using f-k

processing are differently sensitive for NCC and ICC at frequencies below 5 Hz (Fig. 3c versus Fig. B2b) as ICC have limited energy at low frequency (Fig. 4a). Given the sensitivity kernels for Rayleigh wave phase velocity (Fig. 4b) and the dispersion curves obtained from the cross-correlation sections ((Fig. 3c and B2c), Rayleigh waves that are reconstructed with NCC are


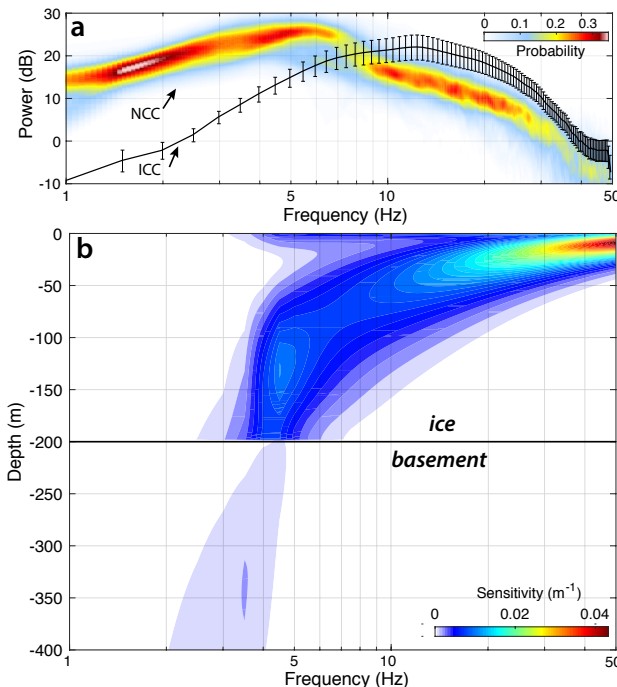

**Figure 4.** (a) Probability density function of noise cross-correlation (NCC) spectra (colors) and median average of icequake cross-correlation (ICC) spectra (black line). Note that raw data (i.e continuous noise or icequake waveforms) were spectrally whitened between 1-50 Hz prior cross-correlation. Due to spectral content of englacial noise and icequakes, NCC and ICC have different depth sensitivity due to spectral response. (b) Sensitivity kernels for phase velocities of the Rayleigh wave fundamental mode for an ice thickness of 200 m over a semi-half space representing the bedrock. The kernels were computed using the freely-available code of Haney and Tsai (2017).

capable of sampling basal ice layers and bedrock while ICC are more accurately sensitive to the ice surface. These results reflect the S-wave velocity dependence on depth.

## 3.2 Dispersion curve inversion and glacier thickness estimation

As discussed above, Rayleigh waves are well suited to determine shear-wave velocities and thicknesses of shallow layers as they are primarily sensitive to the S-wave velocity at a depth of approximately $1/3$ of the wavelength (Song et al., 1989). Sensitivity of Rayleigh waves obtained on NCC to frequencies below 5 Hz enable us to explore the subsurface structure of the glacier with inversions of velocity dispersion curves. Due to the general noise source locations up-flow and down-flow of the network, we limit our analysis to receiver pairs whose accurate GF could be obtained. We thus compute the dispersion curves on eight along-flow receiver lines which constitute the array (inset map in Fig. 5). For each line, we invert the 1D ground profile which best matches seismic velocity measurements in the 3-20 Hz frequency range, using the neighbourhood algorithm encoded in the Geopsy software (Wathelet, 2008).



**Table 1.** Parameter ranges and fixed parameters for grid search to invert the dispersion curves in Fig. 5 for ice thickness. Poisson's ratios of ice and granite were varied between 0.2 and 0.5. Poisson's ratio, ice thickness and P-wave velocity $V_p$ were coupled to the S-wave velocity $V_s$.

| Material | Thickness (m) | $V_p$ (m/s) | $V_s$ (m/s) | Density (kg/m$^3$) |
|---|---|---|---|---|
| Ice | 50-500 | 3870 (fix) | 1500-2100 | 917 (fix) |
| Granite | $\infty$ | 3000-6000 | 1000-3500 | 2750 (fix) |

Following Walter et al. (2015), we assume a two-layer medium consisting of ice and underlying granite bedrock. This is a
simplified approximation and does not include lateral or longitudinal variations in basal topography along the receiver line, nor allow for the presence of snow and basal till layers as well as other heterogeneities such as cavities, crevasses and other water-filled channels. The grid search boundaries for seismic velocity, ice thickness, density and Poisson's ratio are given in Table 1. We fix the seismic P-velocity in ice to 3870 m/s as measured in Fig. 3b, and couple all varying parameters to the S-wave velocity structure.

Figure 5 a-b show the inversion results for the receiver line at the center of the array labelled "4". Velocity measurements are indicated by yellow squares and dispersion curves corresponding to explored velocity models are in colors sorted by misfit values. Walter et al. (2015) explored the sensitivity of the basal layer depth to the other model parameters and report a trade-off leading to an increase in inverted ice thickness when increasing both ice and bedrock velocities. In our inversion, the S-wave velocity is well resolved in the ice layer as all best matching models yield to $V_s = 1707$ m/s. The best fitting model gives an
ice thickness of 236 m and bedrock S-velocity of 2517 m/s. A 100 m/s increase in the bedrock S-velocity results in an increase in ice thickness up to 15 m. Also 3D effects could lead to some errors in the depth inversion results which need to be further investigated.

From the eight receiver line inversions, we find average S-wave velocities of 1710 m/s for the ice, 2570 m/s for the granite and a P-wave velocity of 4850 m/s in the basement, which is consistent with our measurement for refracted P-waves in Fig.
3b. $V_p/V_s$ ratios are found to be 2.2 and 1.9 for ice and granite, giving Poisson's ratios of 0.37 and 0.3, respectively. For the receiver lines near the array edges (Lines 1-3 and 8), the inversion yields to a top layer of thickness 15 m and 7 m respectively, where P-wave velocity is slightly decreased while S-wave velocity is maintained to $\sim 1710$ m/s as for the ice. In this thin top layer, we find a $V_p/V_s$ ratio around 1.6 which corresponds to a Poisson's ratio of 0.2. This is what is expected for snow although only a $\sim 5$ m snow cover was present in the area at the time of the experiment. This top layer could also at least partially be
attributed to the presence of pronounced crevasses near the glacier margins, which do not extend deeper than a few dozens of meters (Van der Veen, 1998).

Inversion results for the ice thickness are plotted in red in Fig. 5c. Associated uncertainties are given by the models which fit the dispersion curves with misfit values below one standard deviation of all 2500 best-inverted models. The black solid line shows the across-flow profile of the glacier baseline, that was averaged over the nodal sensor positions and extracted from the
DEM of Glacier d'Argentière (Sect.2.2.1). Results show that ambient noise interferometry determines the depth of the basal



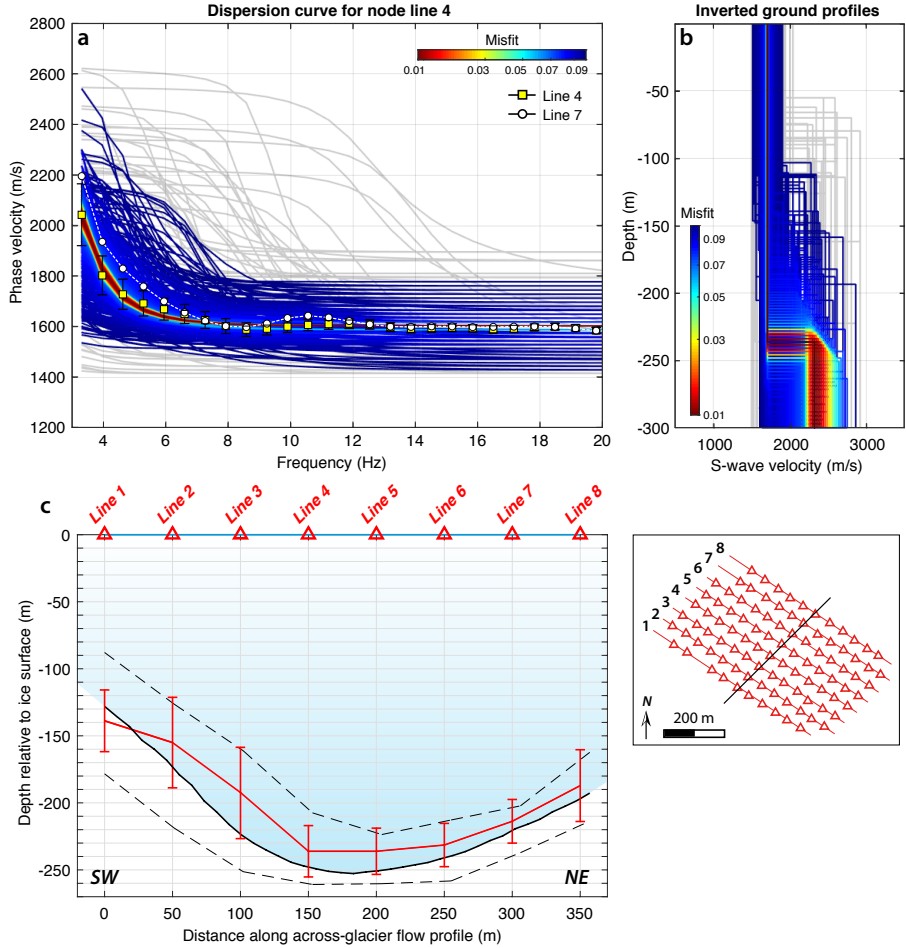

**Figure 5.** Inversion of glacier thickness using velocity dispersion curves of Rayleigh waves uand the Geopsy neighborhood algorithm. Dispersion curve measurements are obtained from f-k analysis of noise cross-correlations on eight receiver lines whose geometry is described in the bottom-right panel. (a-b) Colour-coded population of (a) dispersion curve fits and (b) S-wave velocity profiles for the node along-flow line labelled 4 in (c). Warmer colours correspond to smaller misfit, gray lines correspond to models with misfit values higher than 0.1. In (a) the dispersion curve and uncertainties obtained from seismic measurements are overlaid in yellow squares. For comparison, the dispersion curve computed for the node line 7 and associated to a thinner ice layer is plotted in white dots. (c) Across-flow profile of (red line) ice thickness estimates from Rayleigh wave velocities obtained at eight along-flow node lines and (black line) average basal topography from a DEM. Uncertainties in ice thickness estimates (error bars) correspond to seismic inversion results which yield to a misfit lower than one standard deviation of all generated misfit values. Black dashed lines indicate deviations to the glacier baseline around each node line due to longitudinal topography gradients.

interface with a 10 m resolution as we are able to reproduce the transverse variations of the ice thickness. Differences in ice





thickness absolute values between our measurements and the DEM are generally less than 20 m (being the DEM resolution), and the maximum error is 35 m for Line 3.

Errors and uncertainties (red error bars) are primarily linked to bedrock velocities. Potentially, the bed properties can be refined using additional measurements from refracted P-waves that should be reconstructed on NCC obtained on such a dense and large array and stacked over longer times. Ice thickness estimation is also affected by any lateral and longitudinal variations in the subsurface as phase velocities are here averaged over multiple receiver pairs. The confidence interval we obtain for basal depth is similar order to the variations in glacier thickness along the receiver lines due to longitudinal basal topography gradients (black dashed lines). More accurate 3D seismic models of the glacier subsurface could be obtained using additional station pairs as discussed in section 6.

### 3.3 Azimuthal anisotropy from average phase velocities

Smith and Dahlen (1973) show that for a slightly anisotropic medium the velocity of surface waves varies in $2\phi$-azimuthal dependence according to

$$c(\phi) = c_0 + A\cos\left[2(\phi - \psi)\right] \tag{1}$$

where $c_0$ is the isotropic component of the phase velocity, $A$ is the amplitude of anisotropy and $\psi$ defines the orientation of the anisotropic fast-axis. Several studies have been able to measure azimuthally-dependent phase speed anisotropy from ambient noise correlations computed on dense arrays, and relate them to the presence of geological features and cracks in the Earth's crust or temperature variations in the upper mantle (e.g. Fry et al., 2010; Lin et al., 2011; Mordret et al., 2013).

On glaciers, azimuthal anisotropy can be induced by englacial crevasses, with fast direction for Rayleigh wave propagation being expected to be oriented parallel to the crack alignment (Lindner et al., 2018a). Anisotropy in glaciers and ice-sheets can also result from preferred orientations of ice crystals. There exists different ice fabrics and each one represents a transversely isotropic medium with different types of symmetry (Diez and Eisen, 2015; Maurel et al., 2015). In particular, girdle fabrics can have horizontal axes of symmetry and also induce azimuthal anisotropy as observed by Smith et al. (2017) in polar ice.

The dense array experiment of Glacier d'Argentière covers a wide range of azimuths $\phi$ defined by the orientation of the station pairs, which allows us to investigate azimuthal variation in Rayleigh wave velocities at any given sensor. In order to cover a maximum range of $\phi$-azimuth, we compute velocity dispersion curves for Rayleigh waves obtained on the correlation functions computed on icequake signals (ICC), since correct GF estimates from ambient noise are limited to station pair directions with azimuth $\phi$ roughly aligned with ice flow ($\phi \sim 120^o$, Sect. 3.1). To measure phase velocities at each station path and at different frequencies, we use a slant-stack technique similar to Walter et al. (2015), to octave-wide frequency ranges by bandpass filtering the individual ICC (Appendix B3). We investigate five discrete frequencies in the range 15-30 Hz. The lower frequency bound (15 Hz) arises from the longest wavelength we can resolve given minimum sensor spacing (see Appendix method section B).

For each sensor position, we obtain $c$ velocity measurements as a function of the $\phi$-azimuth of the receiver pair that includes the target station (Fig. 6a). The anisotropy parameters $\psi$ and $A$ are then computed by fitting $c(\phi)$ with equation 1 using a Monte-



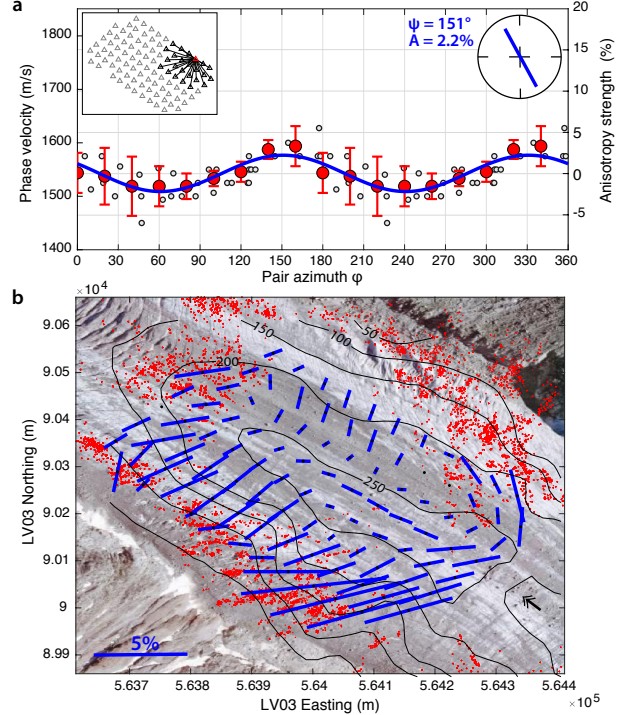

**Figure 6.** (a) Azimuthal variation of phase velocities measured at one node (red triangle in the inset map). White dots are the phase velocity measurements obtained for different azimuths $\phi$ that are defined by the station pair orientation. To avoid spatial averaging, we only consider subarrays of 250 m aperture around the target node at the center as described by black triangles in the inset map. Red dots are phase velocities averaged in 20° azimuth bins. The thick blue curve is the best fit for the $2\phi$ azimuthal variation of the averaged velocity measurements in red. Fast-axis angle and anisotropy strength are indicated in the top right corner circle. (b) Map of fast-axis direction and amplitude of anisotropy measured at 25 Hz. Locations of icequakes active for 7 days are plotted in red dots to highlight the orientations of surface crevasses. Basal topography contour lines are indicated every 50 m. Black arrow indicates ice flow direction. Photo source: Federal Office of Topography © swisstopo.

Carlo inversion scheme. Note that the derivation of Equation 1 also gives rise to an additional $4\phi$ dependence of velocities. Lindner et al. (2018a) used a beamforming approach on icequake records at 100 m-aperture arrays and found that adding the $4\phi$ component to describe the azimuthal variations of phase velocities induced by glacier crevasses yields similar $\psi$ and $A$. We therefore neglect the $4\phi$ term in the present analysis. Finally, to reduce the effect of spatial averaging, we compute anisotropy parameters considering subarrays of stations that lie within 250 m of the target point (inset map in Fig. 6a).

Anisotropy is observed to be more pronounced near the glacier margins (lines 1-2 and 8 as labelled in Fig. 5c), where the anisotropy strength varies between 2% and 8% (Fig. 6b). There, fast-axis directions of Rayleigh wave propagation coincide with the observed surface strike of the ice-marginal narrow crevasses that are also responsible for the generation of icequakes





indicated by red dots. At other locations, fast-axis directions indicate the presence of transversal crevasses with weaker degrees of anisotropy up to 4%. While the near-surface crevasses observed at the array edges result from shear stress from the margin
of the glacier, the transversal crevasses are formed by longitudinal compressing stress from lateral extension of the ice away from the valley side walls, which is typical for glacier flow dynamics in ablating areas (Nye, 1952).

Alignment of the fast-axis directions with that of ice flow appears along the central lines of the glacier (receiver lines 4-5) with anisotropy degrees of 0.5% to 1.5%. This feature is only observed along the deepest part of the glacier where it flows over a basal depression. Results are here computed for seismic measurements at 25 Hz and maps of anisotropy do not change
significantly with frequency over the 15-30 Hz range. Given sensitivity kernels for Rayleigh waves (Fig. 4b), frequencies above 15 Hz are most sensitive to the top 100 m layer. However we do not exclude the possibility that such seismic measurements could be averaged over the entire ice column due to the large spectral band used for velocity estimates. Indeed, we extend our analysis down to 7 Hz and notice that the aligned-flow fast-axis pattern starts to become visible at 10 Hz. The octave-wide frequency range used for computing the velocity at 10 Hz gives corner frequencies of 6-13 Hz for ICC filtering and then include
Rayleigh waves that are sensitive to basal layers.

Generally, we observe an increase in the degree of anisotropy with frequency, which is evident for a shallow anisotropic layer. Conversely increase in anisotropy strength at lower frequency would indicate deeper anisotropic layer. At the Alpine plateau Glacier de la Plaine Morte, Lindner et al. (2018a) find azimuthal anisotropy at frequencies 15-30 Hz with strength up to 8%. They also find that constraining the depth of the anisotropy layer is not straightforward as there exists a trade-off
between its thickness and the degree of anisotropy. Without any further modelling effort, we refrain from further interpreting our results in terms of crevasse extent and depth of the anisotropic layer or any other cause for the observed anisotropic patterns.

## 4 Matched-field processing of englacial ambient seismic noise using the GIS array

As pointed out earlier, localized englacial noise sources related to water drainage can prevent the reconstruction of stable GF. In addition, noise sources change over time preventing stable GF reconstruction when averaged over short-time scale (Preiswerk
and Walter, 2018). These kind of strong, localized seismic noise sources within a seismic array create spurious arrivals in the NCC, preventing GF convergence (i.e. Walter, 2009; Zhan et al., 2013; Preiswerk and Walter, 2018). In this case, the workflow processing traditionally used in NCC procedure as presented in Bensen et al. (2007) and Sect. 3 is not sufficient. Accordingly, we need to apply more advanced processing methods that can reduce the influence of localized sources, and enhance a more even distribution of the noise sources around receiver pairs.
One of the approaches we here apply is Matched-Field Processing (MFP) (Kuperman and Turek, 1997). MFP was used for location and separation of different noise sources in various applications, i.e., to monitor geyser activity (Cros et al., 2011; Vandemeulebrouck et al., 2013), in an exploration context (Chmiel et al., 2016), in geothermal field (Wang et al., 2012) and fault zone (Gradon et al., 2019) event detection. MFP was also used by Walter et al. (2015) to measure phase velocities of moulin tremor signals on the GIS.





Moreover, joint use of MFP and the Singular Value Decomposition (SVD) of the cross-spectral density matrix allows separation of different noise source contributions, as in Multi-Rate Adaptive Beamforming (MRABF: Cox (2000)). The SVD approach was explored by i.e. Corciulo et al. (2012) to locate weak amplitude subsurface sources, and Chmiel et al. (2015) for microseismic data denoising. Also, Seydoux et al. (2017) and Moreau et al. (2017) showed that the SVD-based approach improves the convergence of NCC towards empirical GF. Here, we combine MFP and SVD in order to remove spurious arrivals
in NCC caused by the moulin located within the GIS array, and thus improve the GF emergence.

### 4.1     Location of dominant noise sources via matched-field processing

Röösli et al. (2014) and Walter et al. (2015) documented the presence of hour-long tremor signals in GIS seismic records, typically starting in the afternoon hours. These events occurred on 29 days out of the 45-day total monitoring period. Signal intensity and duration depended on the days of observations, and the energy was mostly concentrated in the 2-10 Hz range
within distinct frequency bands (Fig. 1b). Röösli et al. (2014) and Röösli et al. (2016b) showed a clear correlation between water level in the moulin, and start and end times of the tremor, therefore the tremor signal is referred to as "moulin tremor". Figure 1c shows a spectrogram of a moulin tremor lasting for 10 hours on the night of 28th/29th July 2011 and recorded at one station located 600 m away from the moulin. This signal is generated by the water resonance in the moulin, is coherent over the entire array, and dominates the ambient noise wavefield during peak melt hours (Röösli et al., 2014; Walter et al., 2015).

We briefly summarise the basics of MFP and the details of the method can be found in Cros et al. (2011); Walter et al. (2015) and Chmiel et al. (2016). MFP exploits the phase coherence of seismic signals recorded across an array. It is based on the match between the Cross-Spectral Density Matrix (CSDM) and a modelled GF. The CSDM captures the relative phase difference between the sensors, as it is the frequency-domain equivalent of the time-domain correlation of the recorded data. The CSDM is a square matrix with a size equivalent to the number ($N = 13$ for our GIS array) of stations ($N$-by-$N$ matrix).
MFP is a forward propagation process. It places a "trial source" at each point of a search grid, computing the model-based GF on the receiver array, and then calculating the phase match between the frequency-domain-modeled GF and the Fourier transform of time-windowed data. The optimal noise source location is revealed by the grid points with maximum signal coherence across the array.

In order to calculate the MFP output we use 24 h data of continuous recordings on the $27^{\text{th}}$ of July which encompass the
moulin tremor. We calculate a daily estimate of the CSDM by using 5 min-long time segments in the frequency band of 2.5 and 6 Hz, which gives in total $M = 288$ of segments for one day. This ensures a robust, full-rank estimation of the CSDM ($M >> N$). The modelled GFs are computed over the two horizontal spatial components (Easting and Northing) using a previously optimized Rayleigh wave velocity of $c = 1680$ m/s corresponding to the propagation of Rayleigh waves within the array obtained by Walter et al. (2015). The MFP output is averaged over 30 discrete frequencies in the 2.5-6 Hz range.

The lower frequency bound (2.5 Hz) ensures a higher rank regime of the seismic wavefield, as defined in Seydoux et al. (2017). It means that the degree of freedom of the seismic wavefield is higher than the number of stations. The degree of freedom of the seismic wavefield is defined as a number of independent parameters that can be used to describe the wavefield in the chosen basis of functions. This number depends on the analyzed frequency, slowness of the medium (inverse of velocity),



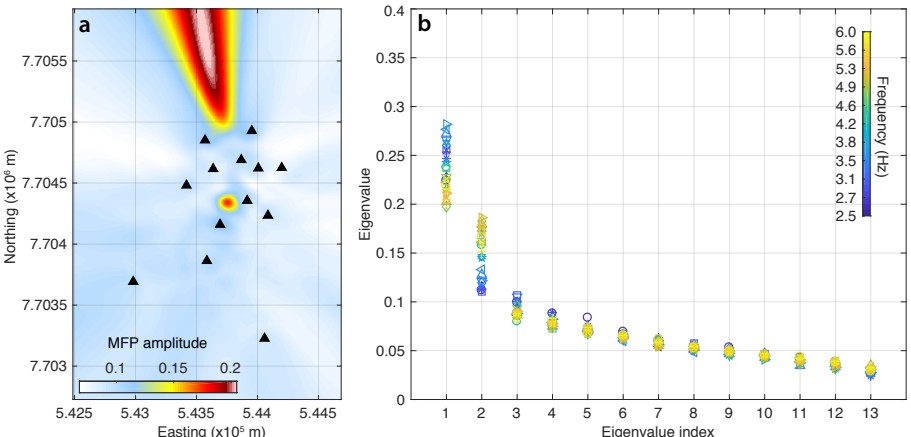

**Figure 7.** (a) Location of the dominant noise sources using MFP in the frequency band of 2.5 Hz and 6 Hz (the MFP output is averaged over 30 discrete frequencies). The MFP was calculated using daily data recorded on the 27th of July at the 13 presented stations (black triangles).(b) Eigenvalue distribution of the CSDM for 30 discrete frequencies in the analyzed frequency band.

and the average interstation spacing of the array (here 736 m). The higher frequency bound (6 Hz) ensures no spatial aliasing

in the beamformer output, given the minimum sensor spacing of 156 m.

Figure 7a shows the grid search for MFP performed over Easting and Northing positions. In order to reveal the location of the source, we use the conventional Bartlett processor (Baggeroer et al., 1993) to measure the match between the recorded and modelled wavefield. The MFP output reveals two dominant noise sources: a well constrained focal spot corresponding to the moulin position inside the GrIS array and another source located North of the array. The latter source is revealed by a

hyperbolic shape. This shape is related to a poor radial resolution of the beamformer for sources located outside of an array. Walter et al. (2015) suggested that this dominant source might correspond to another moulin as satellite imagery shows the presence of several drainage features North of the array. Both noise source signals contribute to the NCC. However, while the source located outside of the array contributes to the stationary-phase zone ("end-fire lobes") of certain receiver pairs, the moulin located within the array will mostly cause spurious arrivals on NCC. In order to separate the contribution of these

noise sources we first perform SVD of the CSDM, and then use a selection of eigencomponents and eigenspectral equalization (Seydoux et al., 2017) to improve the convergence of NCC towards empirical GF.

## 4.2  Green's function emergence from eigenspectral equalization

SVD is a decomposition of the CSDM that projects the maximum signal energy into as few coefficients as possible (i.e. Moonen et al., 1992; Konda and Nakamura, 2009; Sadek, 2012). It allows to split the recorded wavefield into a set of linearly

independent eigencomponents, each of them corresponding to the principal direction of incoming coherent energy and bearing



their own seismic energy contribution:

$$K = USV^T = \sum_{i=1}^{N} K_i = \sum_{i=1}^{N} U_i \Gamma_i V_i^T \tag{2}$$

where, $K$ is the CSDM, $N$ is the number of receivers, $U$ and $V$ are orthogonal matrices containing the eigenvectors, and $S$ is a diagonal matrix representing the singular $\Gamma$ values, and $^T$ denotes the transpose of the matrix. The total number of eigenvalues corresponds to the number $N$ of receivers. The CSDM can be represented as the arithmetic mean of individual CSDMs ($K_i$), where each $K_i$ is a CSDM corresponding to a given singular value $\Gamma_i$.

The SVD separates the wavefield into dominant (coherent) and subdominant (incoherent) subspaces. It has been shown that the incoherent sources correspond to the smallest singular values (Bienvenu and Kopp, 1980; Wax and Kailath, 1985; Gerstoft et al., 2012; Seydoux et al., 2016, 2017). Therefore, a common noise removal method consists of setting a threshold that distinguishes between coherent signal and noise, and to keep only the eigenvectors that are above the threshold before reconstructing the CSDM (Moreau et al., 2017). The CSDM reconstruction consists of eigenspectral normalization (as explained in the following) and summing a selection of individual CSDMs ($K_i$). The "denoised" NCC in the time domain are obtained with the inverse Fourier transform of the reconstructed CSDM.

Here, we follow the approach of Seydoux et al. (2016) for choosing the threshold. In the 2.5-6 Hz frequency band, the wavefield is undersampled by the seismic array (which means that the typical radius of GrIS seismic array is smaller than half a wavelength of the analyzed Rayleigh waves). Seydoux et al. (2016) showed that in this case, the eigenvalue cut-off threshold should be set to $N/2$ in order to maximize the reconstruction of the CSDM. This means that we reject the last eigenvectors (from $7^{\text{th}}$ to $13^{\text{th}}$) as they do not contain coherent phase information.

Figure 7b shows the eigenvalue distribution for 30 discrete frequencies in the analyzed frequency band. The first two eigenvalues correspond to the two dominant noise sources visible on Fig. 7b, and show larger value variation with frequency in comparison with the rest of the distribution. This might be related to the change in the distribution of the dominant sources depending on the frequency. Moreover, the SVD decays steadily, and does not vanish with high eigenvalue indexes. This confirms that the number of the degrees of freedom of the ambient wavefield is higher than the number of receivers.

The CSDM can be then reconstructed by using only individual eigenvectors as in:

$$K_i = U_i V_i^T \tag{3}$$

Note that we do not include the eigenvalues $\Gamma$ in the CSDM reconstruction, which is equivalent to equalizing them to 1. That is why we refer to the reconstructed CSDM as "equalized" (Seydoux et al., 2016).

Figure 8 shows the equalized and reconstructed CSDM by using individual eigenvectors that correspond to the principal directions of incoming coherent energy and are related to different noise sources. Each plot represents MFP grid-search output computed on reconstructed CSDM. Figure 8b shows that the second eigenvector corresponds to the moulin source located inside the array. However, we note that the first eigencomponent also reveals a weaker focal spot corresponding to the moulin location. This indicates, similarly to the singular value distribution, that the spatial distribution of dominant noise sources varies

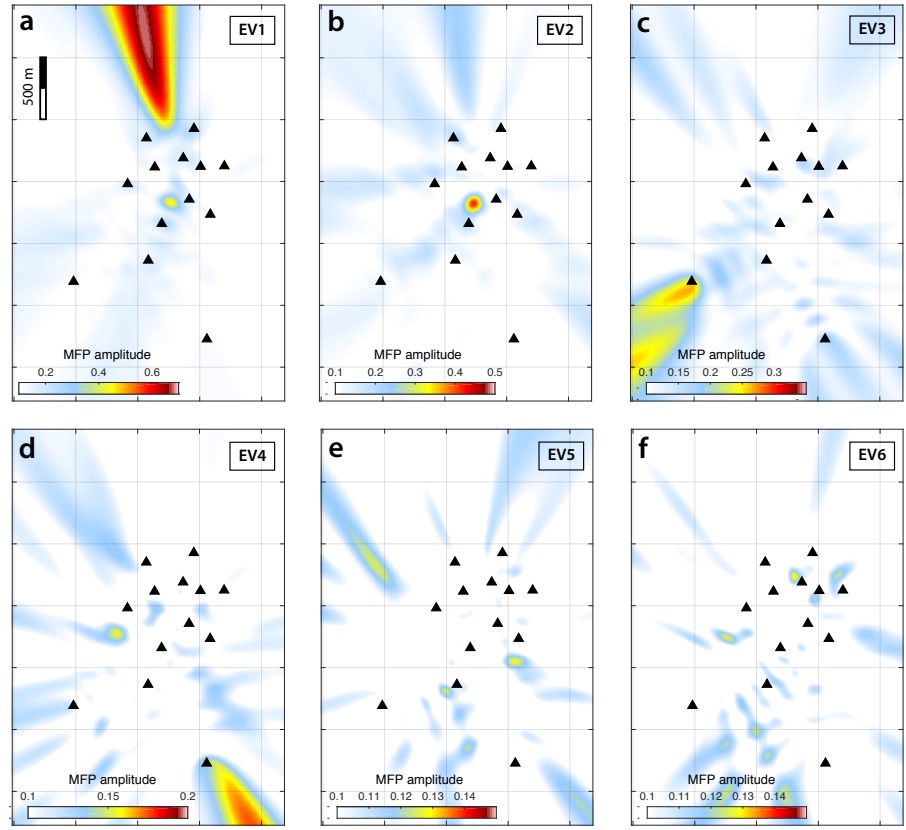

**Figure 8.** Reconstruction of the CSDM by using individual eigenvectors that are related to different noise sources. Each plot shows MFP output computed using the reconstructed CSDM with individual eigenvectors as in equation 3. Each figure represents the MFP gridsearch output calculated for the first eigenvectors: (a) 1, (b) 2, (c) 3, (d) 4, (e) 5, and (f) 6. The spatial coordinates are the same as on Fig. 7a.

with frequency. Furthermore, higher eigenvectors do not reveal any strong noise sources localized within the array, and their MFP output points towards sources located outside of the array.

This MFP-based analysis of spatial noise source distribution allows us to select the eigenvectors of CSDM that contribute to noise sources located in the stationary phase zone (i.e. in the endfire lobes of each station path). We now reconstruct the NCC in the frequency band of 2.5-6 Hz with a step equivalent to the frequency sampling divided by the number of samples in the time window (here 0.0981Hz, so 1019 individual frequencies in total). We perform the inverse Fourier transform of the equalized CSDM reconstructed using the $1^{\text{st}}$, $3^{\text{rd}}$, $4^{\text{th}}$, $5^{\text{th}}$, and $6^{\text{th}}$ eigenvectors. Note that we do not include the eigenvalues

$\Gamma$ in the CSDM reconstruction, which is equivalent to equalizing them to 1. This is why we refer to the reconstructed CSDM as "equalized" (Seydoux et al., 2016).

    Figures 9a and 9b compare the NCC before (in a) and after (in b) the eigenvector selection and eigenspectrum equalization procedure. The displayed NCC are bin-averaged in fixed distance intervals (every 100 m) in order to improve the signal-to-





noise ratio (SNR). The blue line shows the propagation of the Rayleigh waves with the velocity of 1680 m/s. On 9a we observe
spurious arrivals (marked with green dots) that dominates the NCC together with a non-symmetrical shape. On average, the
CSDM equalization process (Fig. 9b) enhances the symmetry of NCC by 40%. To quantify the symmetry of NCC we used the
correlation asymmetry as proposed in Ermert et al. (2015), equation 11.

Unfortunately, we notice that the equalization process reduces the overall SNR ratio and does not eliminate all spurious
arrivals. This might be related to the imperfect separation of different noise sources. For example, we still keep some contri-
bution of the central moulin in the $1^{st}$ eigencomponent. Moreover, by removing the second eigenvalue we remove not only the
seismic signature of the moulin, but also contribution of coherent far-field sources.

To further assess the isotropy of the reconstructed noise field, we use the conventional plane wave beamformer (e.g. Veen and
Buckley, 1988). The plane wave beamforming technique estimates the isotropie and coherence of the ambient seismic noise
wavefield with respect to the slowness and back-azimuth. For the plane wave beamforming calculation, we use the original
(9c) and the previously equalized CSDM (9d). Figure 9c, d shows the beamforming output before (in c) and after the selection
and equalization of eigencomponents (in d). The wavenumber vectors are normalized by the wavenumber corresponding to
Rayleigh wave slowness of $s = 1/1680$ s.m$^{-1}$. A perfectly isotropic noise wavefield consisting of Rayleigh waves would
locate energy near the slowness circle of radius 1. After the removal of the second eigenvalue and the equalization of the
strongest eigenvectors, we observe a more isotropic wavefield, meaning other noise sources are enhanced. This quasi-circular
shape reflects the energy that arrives from different azimuths. The difference in beamformer amplitude can be caused by the
non-regular shape of the GrIS array and different energy contributions of the noise sources. The results show not only the strong
source of noise coming from the North, but also energy incident from the South-West that might be related to oceanic ambient
noise in the Labrador Sea (Sergeant et al., 2013) or other continuous noise generated by calving and ice-mélange dynamics
in the proglacial fjord of nearly Jakobshavn Isbræ (Amundson et al., 2010), one of Greenland's largest ice-streams. Finally, it
seems that not much seismic energy is incident from the inland of the East GIS.

After the eigenspectrum equalization, we are able to extract Rayleigh waves dispersion curve from the averaged seismic
section obtained in Fig. 9b. For calculating the averaged dispersion curve we use a version of the Aki's spectral method
(Aki, 1957) which consists of fitting a Bessel function to the real part of the cross-correlation spectrum. This method is
referred to as SPAC and is described in Appendix B2. The Rayleigh wave phase velocity dispersion curve averaged over all
station measurements is shown in Fig. 9e with yellow squares and errorbars representing the measurement discrepancies for
individual NCC. The dotted line presents an apriori Rayleigh velocity dispersion curve extracted from Walter et al. (2015).
High discrepancies observed at lower frequencies mainly arise from the limited frequency band for computing the NCC (see
the appendix method section B). The slight differences between the two dispersion curves might be related to the different
approaches used for phase velocity dispersion curve extraction (MFP in Walter et al. (2015), and SPAC in the current work).
Moreover, Walter et al. (2015) worked on a wider frequency band and averaged their dispersion measurements over 46 days
and in our study we use only one-day of data.

Several additional tests could be used to further improve the SNR of the NCC and their convergence to GF. A similar
procedure could be performed for other days. It would be useful to find an automatic criterion of the eigenvalue selection based



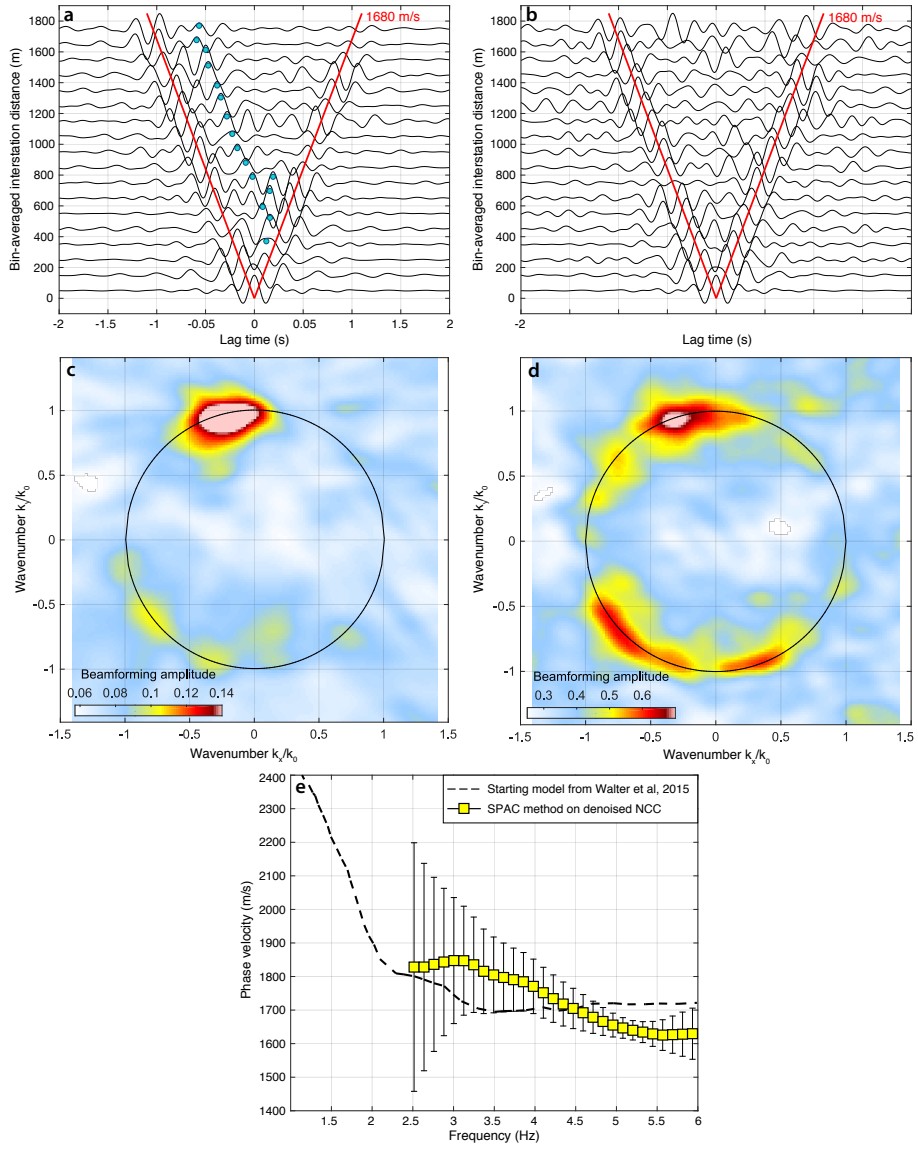

**Figure 9.** (a) Stacked sections of NCC in the frequency 2.5 to 6 Hz. The red line show the propagation of the Rayleigh waves with velocity of 1680 m/s (also in b). Spurious arrivals are marked with green dots. (b) Stacked sections of NCC reconstructed in the frequency 2.5 to 6 Hz from the CSDM eigenspectrum equalization. (c) Plane wave beamforming before the eigenspectrum separation and normalization, and (d) afterwards. (e) Rayleigh wave phase velocity dispersion curves (yellow squares) calculated over the averaged seismic section in (d) with errorbars indicating discrepancies in velocity measurements at different station paths. The dispersion curve is obtained with the Aki's spectral method. Dashed black line is the dispersion curve obtained by Walter et al. (2015) with MFP.





on the near-field beamforming (calculated over spatial positions with no plane wave approximation, as in Fig. 7a) and far-field
beamforming (calculated over wavenumbers using the the plane wave approximation, as in Fig. 9c, d). Another improvement
could consist of azimuthal stacking the NCC according to the direction of the noise sources, although the GrIS array does not
have sufficient azimuthal and spatial coverage to implement this. Moreover, we could envisage calculating a projector based
on the SVD (as in MRABF) only for the time period when the moulin is active and then project-out the moulin signature from
the continuous seismic data.

In summary, we conclude that the CSDM eigenspectrum equalization together with beamforming-based selection of eigen-
vectors is an useful method to separate localized seismic sources in a glaciated environment. It can further improve the GF
emergence from ambient seismic noise in the presence of strong, localized englacial noise sources for imaging applications.

## 5 Cross-correlation of icequake coda waves using the Gornergletscher array: a window-optimization approach

Inhomogeneities in the Earth's crust give rise to coda waves (following the direct seismic arrivals after an earthquake) consisting
of reflections and multiply scattered waves (Aki and Chouet, 1975). The same effect can provide an isotropic wavefield to allow
for GF retrieval via the cross-correlation of a coda window at two sensors (Campillo and Paul, 2003), as multiply scaterring
should create a homogeneous illuminating pattern (Shapiro et al., 2000; Hennino et al., 2001).

In contrast, glacier ice is highly homogeneous with scattering crevasses mostly confined to the surface crevasse zone, typi-
cally 30 m-thick (Van der Veen, 1998; Podolskiy and Walter, 2016). As a result, icequakes typically lack a strong coda (when
compared to earthquakes, Fig.1a), usually limited to a few milliseconds after the ballistic waves, with a fast decay of seismic
amplitudes toward the noise level. However, focusing on few strongest events recorded in Gornergletscher (Fig. 2c), we still
observe sustained coda (1.5 s-long) above the background noise (Fig. 1c) whose origin is discussed below. In the following,
we explore the application of coda wave interferometry (CWI) on selected near-surface icequakes to retrieve the GF, which
could not be obtained from traditional processing of icequake cross-correlations (as defined in Walter et al. (2015) and used at
Argentière array) because of lacking sources in the stationary phase zones of the seismic array (Sect. 3). The use of icequakes
here is fundamentally different than in section 3, in that the ballistic arrivals are specifically avoided (whereas in the other case,
the ballistic component was the primary source of energy in the cross-correlation functions).

### 5.1 Origin of coda waves

The coda of a local earthquake is a mix of multipath arrivals and is formed by waves that are deviated from their original path by
multiscale heterogeneities (i.e. velocity or density anomalies as cracks, rock pores, fluid bubbles, etc) present in the propagation
medium (Aki and Chouet, 1975). The seismic coda can also consist of reverberations in horizontally layered structure under
the receiver, reverberations in layered structures between the source and receiver (Bouchon, 1982), and surface waves scattered
by lateral heterogeneities (Aki, 1969). In addition, coda can be generated by the conversion of body waves into surface waves
by topography at the free surface or at buried interfaces at depth (Spudich and Bostwick, 1987).





In a strong scattering medium, ballistic and single-scattered waves will follow a random walk between the scatterers until
forming a diffusive halo which consists of multiply-scattered waves, and whose propagation velocity is influenced by the
scattering strength often measured by the scattering mean free path (De Rosny and Roux, 2001). The diffuse character of coda
waves has been demonstrated by the observation of the stabilization of the S-to-P energy ratio (Campillo et al., 1999; Shapiro
et al., 2000; Hennino et al., 2001). Contrary to ballistic waves, fully diffuse wavefields are expected to contain all possible
modes and propagation directions following an equipartition principle (Paul et al., 2005; Colombi et al., 2014). Diffusion of
seismic coda waves is the pre-requesite condition for a homogenized seismic wavefield and the correct estimation of the GF
through the application of CWI (Campillo and Paul, 2003; Malcolm et al., 2004; Paul et al., 2005; Gouédard et al., 2008b;
Chaput et al., 2015a, b).

    The strongest 720 events chosen out of more 24000 icequakes detected at Gornergletscher exhibit a sustained coda with
approximate duration of 1.5 s (Fig. 1c). The propagation regime of seismic waves can be identified by the evolution of the
elastic intensity ("coda power spectrum") being the squared seismic amplitudes (top panel in Fig. 1c). Before the source energy
has reached the receiver, the elastic intensity is equal to some background or ambient level. Once the source pulse arrives at
the receiver, the intensity rises up and then begins to decay exponentially. This is the ballistic regime. After several mean-free
times, the intensity begins to decay diffusively with time as multiple scattering slows the transport of energy out of the scan
region (Malcolm et al., 2004). This is the diffusion regime and it is characterized by a linear decay of the coda intensity (Aki
and Chouet, 1975). Eventually, intrinsic attenuation (anelastic loss) dominates and the energy falls to the noise level.

    Figure 1c shows such linear decay of the coda power spectrum starting at $\sim 0.5$ s indicating that icequake seismogram signals
contain single or multiply back-scattered energy that may approach from a wide range of directions assuming that the scatterers
are homogeneous around the network site (Chaput et al., 2015a). In the present study we do not investigate further the cause of
wave scattering in glaciers and particularly in Gornergletscher but suggest a relation to the presence of pervasive conspicuous
near-surface crevasses (Fig. 2c) and deeper fractures as intermediate-depth and basal fault planes have been reported at the
study site (Walter et al., 2008, 2009, 2010), as well as topography gradients, reflections at the glacier margins and/or rock and
air inclusions.

## 5.2    Coda wave interferometry and Green's function emergence

We first apply a standardized CWI processing scheme following Gouédard et al. (2008b). The cross-correlations are computed
on 10-30 Hz spectrally whitened seismograms to reduce the influence of background noise. As a first guess, coda waves are
arbitrarily time windowed around 0.5-1 s (see Fig. 1c) by looking at the decay of the waveform amplitudes. The first sample of
the coda correlation window corresponds to the two station average of the time when the seismogram envelope falls below 5%
of the ballistic wave maximum amplitudes. Because of the decrease of coda amplitude with time, we cannot perform a simple
cross-correlation between the coda signals without strongly overweighing the earliest part of the coda. To avoid this problem,
we follow Paul et al. (2005) by disregarding the amplitudes and considering only one-bit signals (see also Sect. 3).

    Figure 10a shows the individual coda wave cross-correlation functions (CWCC) sorted by the azimuth of the source event
relative to the station path. In contrast to conventional ICC (correlations of icequake direct waves) whose Rayleigh wave arrival



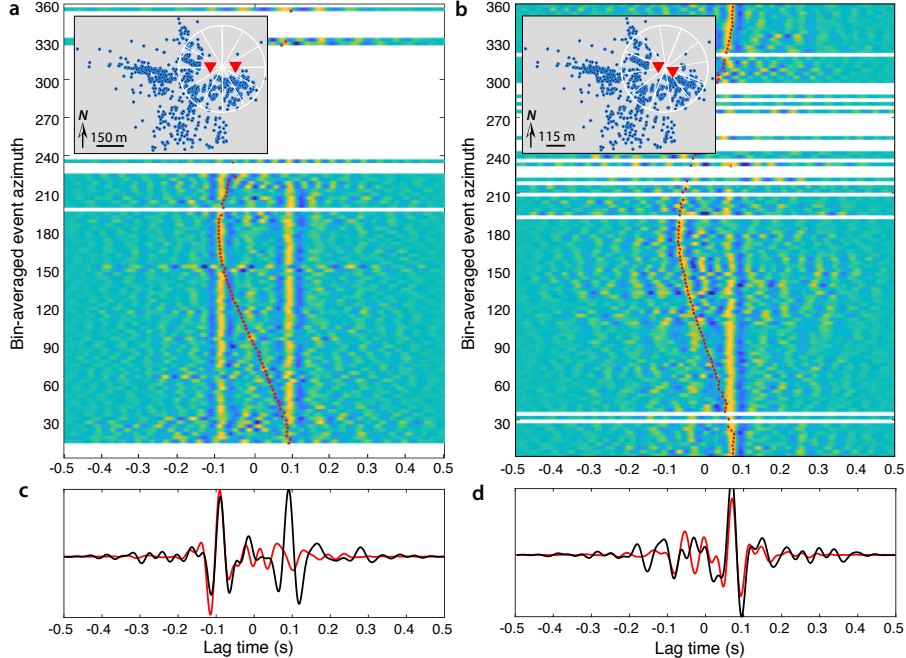

**Figure 10.** (a, b) cross-correlation functions obtained at two station pairs from each source and plotted as a function of event azimuth for a time window in the coda part. Rayleigh wave arrival times of correlation functions obtained for a time window which encompasses direct waves of icequake records are plotted in red dots. Stations, station separation and source geometry are plotted in the inset maps. (c, d) Stack of the correlation functions obtained from icequake direct waves (red) and coda (black).

times (i.e. amplitude maxima of the ICC, see also Fig. B4 and Sect. 3) are plotted in red, the coherent arrival times in the CWCC

no longer depend on the event azimuth. The computed arrival times correspond to stationary Rayleigh waves travelling between the two stations. The causal and acausal parts of the individual CWCC tend to symmetrize as we are in the scattering regime. This results in a symmetric correlation stack (Fig. 10c) whereas only the acausal part of the GF is reconstructed from the ICC due to missing sources behind one of the two stations.

For the pair of closer stations (Fig. 10b), the reconstructed acausal times still depend on the source position while the CWCC

causal times are stable with azimuth. The source position signature on one side of the correlation function could be an effect of heterogenous scatterers which cause single-back scattering and then skew the illumination pattern at one side of the receiver pair. Another explanation could be that the correlation window used here for CWI is still influenced by the incoming energy flux from ballistic waves which then create an anisotropic wavefield (Paul et al., 2005).

Focusing on a complex scattering medium at the glaciated Erebus volcano (Antarctica), Chaput et al. (2015b) showed that

symmetric GF could be recovered when optimizing the icequake coda correlation window over the sources. In the case of a weak scattering medium as glacial ice, the coda time window for the diffusion regime should notably depend on the distance of





the scatterers to the recording seismic sensor. We therefore use a similar optimizing-window processing scheme for improving the GF convergence at each station pair.

The overall processing and technical details of coda window optimization are described in Chaput et al. (2015b). We refer to
the method as MCMC processing as it involves a Markov Chain Monte Carlo scheme. A bayesian inversion determines the best coda window to generate a set of CWCC that are the most coherent and symmetric across the source events. We first construct a matrix of CWCC that are bin-averaged over $N$ events and then iterate this correlation matrix by randomly shifting the coda window along a certain number $M$ of random traces. At each iteration, a misfit function is constructed based on the coherency of the $N$-binned CWCC matrix and the causal-acausal symmetry of the CWCC stack. In the end, the best optimized models
consisting of the cross-correlation matrices computed for different sets of individual event coda windows are stored and used to generate an average stack of CWCC which is our estimate of the final GF.

MCMC processing involves several parameters that need to be tuned:

- the analyzed frequency band (10-30 Hz),

- the number of traces $N$ used to generate event-binned correlation functions (we here use $N = 40$),

- the number of traces $M$ that are randomly selected to be shifted at each iteration ($M = 10$),

- the portion of the cross-correlation stack where we want to optimize the causal-acausal symmetry (i.e. direct waves of the constructed CWCC stack and/or the cross-correlation coda part); we choose here to optimize the CWCC symmetry for both reconstructed direct and coda waves, ie. at later times than what expected for a wave propagating at velocity 1700 m/s,

- the relative importance of the causal-acausal symmetry and the cross-correlation matrix coherency that is used to optimize the misfit function (coefficient factors $A$ and $B$ in Chaput et al., 2015b, , equation 2); we weight the symmetry and coherency evenly,

- the maximum number of iterations we allow, which naturally depend on the number of available sources, $N$ and $M$ (we use up to $2 \times 10^4$ iterations),

- the coda correlation window length $T$; we use $T = 0.5$ s and we force the algorithm to shift the correlation window to no later than 1.5 s in order to stay within the icequake coda and to not correlate noise (see Fig. 1c).

Figure 11a shows the MCMC optimization of the CWCC symmetry at the station pair already presented in Fig. 10d. Blue and red lines are the causal and acausal parts of the correlation stack, respectively. Solid and dashed lines are the resulting CWCC obtained from MCMC processing and standard processing (i.e. first iteration of the MCMC inversion), respectively. While
the direct Rayleigh waves could not be reconstructed in the acausal part of the correlation function using the first coda wave windowing, the MCMC output approaches the symmetrical GF as we see a Rayleigh wave propagation in both directions, and also the emergence of a coherent coda in both parts of the correlation function. MCMC processing also coherently increases the presence of energy at zero lag-time. It is very likely because scattered coda also contains a strong vertically trapped body wave that correlates at 0 across relatively near receivers, even if it is not part of the "true" Green's function. i.e., the wavefield
may be well scattered, given that the MCMC can indeed adequately recover symmetry, but that symmetry may contain modes



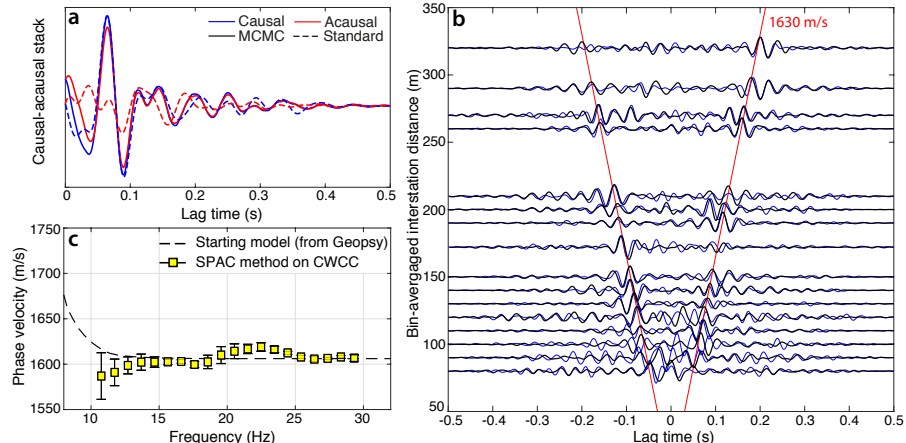

**Figure 11.** (a) Causal-acausal symmetry of the CWCC obtained one station pair using standard coda interferometry processing as in Fig. 10d (dashed lines) and MCMC processing (solid lines). (b) Correlation gather sorted by increasing distance and averaged in 10 m intervals. Black and blue lines result from MCMC and standard processing, respectively. Red lines show the propagation of Rayleigh waves with the velocity of 1630 m/s. (c) Rayleigh wave phase velocity dispersion curve (yellow squares) averaged over measurements for all station pairs. Error bars indicate discrepancies in velocity measurements at different station paths. The dispersion curve is obtained with the Aki's spectral method. Dashed black line is the theoretical dispersion curve for a 160 m-thick ice layer computed with Geopsy.

that are not purely the result of an isotropic point source Green's function. Finally, the MCMC inversion gives optimized coda windows in the range 0.7-1.2 s for the majority of events.

Figure 11b shows the final stack of CWCC gather sorted by increasing inter-station distance and averaged in 10 m-binned intervals. For comparison, CWCC computed with the standard processing are in blue. CWCC are noisier that ICC obtained

from correlations of icequake ballistic waves when computed on homogeneous source distributions (Gouédard et al., 2008b). Nevertheless, at most station pairs, the MCMC processing managed to extract Rayleigh waves with consistent traveltimes in both causal and acausal parts.

We extract a dispersion curve of phase velocities averaged over the station components (Fig. 11c) using the SPAC method already used in section 4.2 (Appendix B2). We find an average Rayleigh wave velocity of 1600 m/s which is in the estimate

range of what Walter et al. (2015) find at Gornergletscher using slant-stacking of ICC arrival times (Appendix B3). Errors introduced at lower frequencies arise from the limited frequency band used for computing the CWCC and filtering effects (see the appendix method section B).

At some locations, the MCMC processing fails in computing symmetric CWCC. First, there still exist differences in the amplitudes of the Rayleigh waves in the causal and acausal parts of the final GF estimate (Fig. 11a), meaning that the icequake

coda is not entirely diffuse and may result from single back-scattering on preferred scatterers in the propagation medium. Secondly, spurious arrivals prior to the expected arrival times for Rayleigh waves likely result from seismic energy arriving from directions which do not point toward the stationary phase zones of the station path. Such spurious arrivals could result





from seismic reflections on the glacier bed beneath the stations, or other glacier noise sources. A certain portion of the icequake coda may be still influenced by background noise sources especially at distant stations from the source as the coda time window
of one station may fall in the noise window of the further one.

Paul et al. (2005) used CWI on regional earthquake seismograms and could not obtain symmetric GF due to anisotropic diffuse wavefields. They argue that the flux of energy from the source can still dominate the late coda and result in nonsymmetric cross-correlations when the distribution of earthquakes is not isotropic around the stations. In the case of Gornergletscher icequakes, we still see the influence of the energy flux approaching from the direction of the source at a few station pairs and
for some events as depicted in Fig. 10b. We try coda wave correlations on the Argentière node grid and notice an influence of the source position on the retrieved CWCC likely because we did not select strong enough signals with sustained coda. Similarly, GF convergence does not work for weak Gornergletscher icequakes.

In general, CWI must be processed on carefully selected events which show sustained coda above the background noise. Moreover, the abundance of seismic sources in glaciers often pollutes coda wave seismograms. We often find the situation
where ballistic body and surface waves generated by repeative and subsequent events (or bed reflections) arrive at the seismic sensor only a few milliseconds after the onset of the first event of interest and therefore fall in its coda window. This typically biases CWCC computation. The brief icequake coda duration and the interevent time distribution impose limitations of CWI on large arrays.

To conclude, even if not perfect, the extraction of GF from coda waves allows imaging a glacier subsurface between station
pairs. In principle, this can be done even in cases when (skewed) distribution of icequake sources or sustained noise sources does not allow for GF estimation.

## 6   Discussion

In this study, we show the possibility for extracting informations on the glacier subsurface structure and elastic properties from cross-correlations of typical seismic wavefields recorded in ablating ice-covered area. The three proposed methods could
theoretically be applied to each one of the explored dataset but were not further tested here. The standard processing schemes proposed in section 3 for computations of NCC and ICC successfully work on Glacier d'Argentière array to estimate accurate GF due to distributed sources. We are able to obtain a precise image of the glacier subsurface (with uncertainties of the order of 10 m) thanks to the dense measurements points which allow to improve the SNR of the cross-correlation functions by stacking them in regular distance intervals. This spatial averaging could be overcome when (1) stacking the GF over longer seismic
time series of typically weeks or months (e.g. Sabra et al., 2005; Larose et al., 2008) or (2) applying more advance processing schemes as the one proposed for GIS and Gornergletscher studies to overcome uneven source distribution effects.

MPF and eigenspectral equalization of the cross-spectral density matrix of noise seismograms (Sect. 4) enable to distinguish propagation directions of incoming seismic energy. This method applied to glacier noise continuous recordings allow to remove seismic signatures from dominant and localized noise sources as in the case of the GIS moulin. It also improves the seismic
wavefield spatial homogenization in order to reveal weaker sources that could be used for extracting the GF at sensor pairs





which initially lacked noise sources in their stationary phase zones. MPF combined to SVD of the cross-spectral matrix could then be applied to Glacier d'Argentière array in order to improve the GF estimates from NCC at station pairs not aligned with the strongest noise sources that lie down and upstream of the array. By doing so, dispersion curves of Rayleigh wave velocities could be obtained at more station pairs and used to invert a more accurate 3D model of the subsurface. The eigenvalue

distribution of the cross-spectral density matrix depends on the number of receivers. The MPF/SVD method is then particularly well suited for large sensor arrays, also the geometry of the glacier discharge system (channelized versus distributed) is the primary controlling factor for successful applications.

In the absence of distributed continuous noise sources, the use of icequakes is a good alternative for GF estimates especially in winter when the glacier freezes and no melt occurs. As icequakes propagate to few hundreds meters, ICC studies are well

suited for medium-size arrays with aperture of typical order of 500 m. In the case of uneven distributions of icequakes which often map the ice pervasive crevasses, the GF estimate can be optimized with CWI and the MCMC approach used in section 5. CWI successfully work on strongest selected events at Gornergletscher as we could measure a seismic coda with amplitudes above noise level. The CWI/MCMC method should be appropriate for ICC studies on smaller icequake datasets (here we show the success of the method when averaged over 700 events) and smaller size arrays which are deployed in targeted regions

where deep crevasses are present, i.e. typically near the glacier margins. As an example, the use of icequake diffuse coda could help on estimating the GF at smaller subarrays at the edges of a larger array as in the configuration of Glacier d'Argentière experiment.

In the following we discuss the type of array deployment, geometry and measurements suitable for structural and monitoring studies using the GF obtained by either one of the processing described above.

## 6.1 Implications for glacier imaging

The capability of extracting accurate seismic velocities on a line of receivers can be substantially improved when stacking the correlation functions on a large number of receiver pairs. Dense array deployment of sensors regularly spaced on a grid enables to densify the measurement points and will thus improves the results. Technological improvements in the portability of seismic instruments and storage capacity make today such kind of experiment feasible to deploy, as it was done on Glacier

d'Argentière.

The performance of an array for deriving phase velocity values in a wavenumber or frequency range depends on its geometry and on the wavefield characteristics (Appendix B). Wathelet et al. (2008) recommend that the array diameter be at least as large as the longest wavelength of interest (we conventionally take two to three wavelengths) and that the station spacing for any direction should be less than half the shortest wavelength of interest to avoid spatial aliasing. To be able to sample small-scale

glacier heterogeneities (e.g. individual crevasses, cavities, short-wavelength basal topography, etc), the investigated seismic wavelength should be of the same order of the anomaly, giving minimum sensor spacing of the order of 10 m. To be able to sample the basal interface and target elastic parameters of sediments constituting the till layer or the bedrock, one should design suitable array geometry. For a glacier thickness of 200 m, we need an array aperture of at least 600 m to measure propagating





surface waves with a one-wavelength cycle. For a 500 m-thick glacier, we need sensors that are at least 1500 m apart, although
prior knowledge on the basal interface allows to better constrain the inversion of depth with lower-size arrays.

Finally, as outlined in Fig. 4, Rayleigh waves reconstructed on the cross-correlations of continuous ambient englacial noise
(NCC) and of icequake records (ICC) are differently sensitive to the subsurface layers, primarily due to the recorded wavefield
characteristics and spectral content. Given typical glacier noise sources, NCC can reconstruct propagating waves down to
frequencies of 1 Hz and thus well sample the basal layer for glaciers thinner than $\sim 600$ m. In contrast, surface crevasse
icequakes have dominant energy at frequencies above 5 Hz. ICC are thus more appropriate to sample the surface layers. To
obtain elastic properties of basal layers in thicker glacier settings and ice-sheets, cross-correlations of oceanic ambient noise
recorded on broadband seismometers should be considered as such sources sample lower frequencies below 1 Hz (Zhan et al.,
2013).

In this study, we only use the vertical data components of the three datasets. Seismic velocity measurements can additionally
be complemented by other types of observations computed on the horizontal and vertical components of the GF, such as the
horizontal-to-vertical (H/V) ratio. This technique can be implemented on cross-correlation functions (Zhan et al., 2013) or on
raw continuous seismograms (Picotti et al., 2017; Preiswerk et al., 2018) as long as the seismic sensors are well oriented and
coupled to the ice. The H/V measurements exploit the ellipticity of the propagating Rayleigh wave which is strongly sensitive
to the vertical-layer structure beneath the recording site. H/V ratio can be computed at different frequencies, we then call it H/V
spectral ratio. The spectrum of the H/V ratio reveals the fundamental SH resonance frequency of a site which is related to the
basal impedance contrast. 1D resonance of S-waves is particularly strong for a "soft" layer (such as ice) on top of a more rigid
layer (such as bedrock). Assuming a horizontally homogeneous medium, the resonance frequency can be used to constrain the
ice thickness (Picotti et al., 2017) or to investigate 3D effects of the recorded wavefield (Preiswerk et al., 2018). It is H/V ratios
and dispersion curves of surface wave velocities can be jointly inverted (Lin et al., 2014) for an even more accurate 3D model
of the glacier subsurface.

## 6.2 Implications for glacier monitoring

Changes in subsurface properties affect seismic wave propagation. Repeated analysis of cross-correlation analysis allows to
detect these changes. The accuracy for detecting and quantifying relative seismic velocity changes $\delta v/v$ from time shifts in the
direct wave traveltime measurements depends on the quality of the GF estimate (i.e. SNR and stability of the GF over time)
and the magnitude of the velocity variation. Seismic velocity monitoring is usually performed in the coda part of the cross-
correlation function through the application of CWI, as multiple scattered coda waves travel larger distances and accumulate
time delays. CWI enables to detect relative velocity changes as small as $\delta v/v \sim 10^{-3}$ (e.g. Sens-Schönfelder and Wegler,
2006; Brenguier et al., 2008a, b; Mainsant et al., 2012). Hadziioannou et al. (2009) further showed that the true GF estimate is
no longer needed for CWI, as long as the noise sources are stable over time.

CWI computed on on-ice seismic recordings could lead to the monitoring of englacial crevasses, failure of calving icebergs,
glacial lake outburst flood, break-off of hanging glaciers, surface mass balance and such bed conditions as the subglacial
hydraulic system and subglacial till porosity and deformation. Such topics are currently investigated by analyzing the emitted





seismicity (e.g. Walter et al., 2008; Roux et al., 2010; Bartholomaus et al., 2015; Preiswerk et al., 2016, 2018; Podolskiy et al., 2017; Lipovsky et al., 2019). Passive seismic monitoring of glaciers could lead to the detection and understanding of processes
related to climate conditions, glacier hydraulics and ice flow dynamics, which today are labor-intensive to investigate with active geophysical measurements or satellite imagery.

Unfortunately, the weak ice scattering limits the emergence of coherent coda in the correlation functions, that are furthermore sensitive to the source distribution (Walter et al., 2015) and can then be affected by the changing nature of primary ambient noise sources (Preiswerk and Walter, 2018). To overcome this source distribution effect, it is possible to design arrays for
multidimensional deconvolution (MDD) of the time-averaged cross-correlations. Seismic interferometry by MDD measures the illumination pattern (e.g. Wapenaar et al., 2011) using recordings by a set of additional receivers along a contour which goes through the virtual-source's location (Fig. 12a). MDD processing then enables to remove the imprint of the (non-uniform) source distribution from the correlation responses and improves the quality of the retrieved GF.

By employing virtual boundary conditions in seismic interferometry by MDD (Weemstra et al., 2017) on active-shot signals,
Lindner et al. (2018b) could create artificial coda following the direct wave. This coda consists of multiple seismic reflections within the (closed) receiver contour trapping the waves. Applied to icequake signals recorded at the Argentière node grid (Fig. 12a), this technique is successful in retrieving stable GF over time with coherent coda energy (Fig. 12b). This approach is a promising candidate for a monitoring technique of any changes in seismic velocities in glaciers even in the absence of scattering coda. Long-term installations directly on or in the ice have increased in recent years due to technological improvements (Aster
and Winberry, 2017). MDD could be applied to glacier seismic sources recorded over time scales longer than a month, using borehole sensors which ensure a solid coupling to the ice.

## 7   Summary and concluding remarks

This study explores the application of seismic interferometry on on-ice recordings to extract the elastic response of the glacier subsurface beneath one array deployment. In contrast to ambient noise studies focusing on the Earth's crust, the GF retrieval
from the cross-correlation of glacier seismicity at sensor pairs is notoriously known as difficult due the limited spatial coverage of glacier point sources and the lack of seismic scattering in homogeneous ice.

We investigate the GF emergence on three particular cases: (A) a favourable distribution of icequakes and noise sources recorded on a dense array (Sect. 3), (B) a spatially limited distribution of icequakes and a dominant noise source constituted by a moulin (Sect. 4), and (C) crevasse-generated scattering (Sect. 5).

In case (A) on Glacier d'Argentière, cross-correlations of ambient noise and icequake recordings result in accurate GF estimates, when averaged in regular inter-station distance intervals. The cross-correlation functions contain P-waves and dispersive Rayleigh waves propagating in the ice. Velocity estimates of seismic waves recorded across the array enable to conduct structural studies, map the glacier bed and englacial crevasses.

In case (B) in GIS, spurious arrival times for Rayleigh waves are obtained from ambient noise cross-correlation and are
attributed to the moulin seismic signature which acts like a secondary dominating source. They can be removed by using

matched-field processing on the cross-spectral matrix of seismic recordings and equalization of eigenspectral values. The processing involves decomposition of the wavefield in independent components of seismic energy contribution. It allows to locate and separate different noise sources and improve the GF convergence by selecting appropriate directional components.

In case (C) in Gornergletscher, cross-correlations of icequake coda waves show evidence for a quasi-homogenized wavefield coming from a wide range of directions as a result of seismic back-scattering likely attributed to pervasive and penetrating englacial crevasses. By optimizing the correlation window in the coda part, we compute symmetric GF which could not be obtained otherwise by cross-correlation of icequake ballistic waves due to a skewed source distribution which creates a non-homogeneous illumination pattern. Coda wave interferometry is a technique usually employed in crustal seismology but works fine on accurately selected strong icequakes with sustained 1.5 s-long coda waves that can be recorded above the background noise level.

Our capacity to extract accurate GF and seismic velocities is much improved using seismic measurements on a dense array as on Glacier d'Argentière. However, matched-field processing and coda wave interferometry allow for new kind of measurements on sparse seismic networks and enable to fasten the GF convergence for non-idealized seismic illumination patterns which arise in homogeneous glacial ice.

Finally, the use of nodal sensor technology enables fast deployment of large N-arrays suitable for seismic interferometry studies. This opens new ways for characterizing and monitoring glacial systems using continuous passive seismic recordings.

*Code and data availability.* All data except for Argentière are available by request on the server of the Swiss Seismological Service (http://seismo.ethz.ch). Obspy Python routines (www.obspy.org) were used to download waveforms and process icequake catalogues. NCC of Argentière dataset were computed using the MSNoise Python package (www.msnoise.org, Lecocq et al., 2014).

*Author contributions.* PR and FG designed the Argentière experiment in the scope of the RESOLVE project (https://resolve.osug.fr/). FL, PR, FW and AS processed the icequake catalogues. AS processed the correlation functions and analyzed the results at Argentière. MFP of Greenland data was processed by MC. CWI at Gornergletscher was processed by AS and FW with the input of JC. Icequake correlations by MDD were processed by FL. AM provided the code for computing phase velocities by fitting the cross-spectrum with a Bessel function. AS and MC prepared the manuscript with contributions from all co-authors.

*Competing interests.* The authors declare that they have no conflict of interest.

*Acknowledgements.* This work was funded by the Swiss National Science Foundation (SNSF) project Glacial Hazard Monitoring with Seismology (GlaHMSeis, grant PP00P2_157551). Additional financial support was provided to AS by the Swiss Federal Institute of Technology





(ETH Zürich). The Observatory of Sciences of the Universe of Grenoble funded the project RESOLVE for the data acquisition at Argentière.
SNSF and ETH Zürich participated to the data collection in Greenland (grants 200021_127197 SNE-ETH and 201 ETH-27 10-3) and on
Gornergletscher (grants 200021-103882/1, 200020-111892/1). We gratefully acknowledge all the people involved in the RESOLVE project
and who participated to the array deployment in Argentière, collected and processed raw formats of seismic data. We thank Olivier Laarman,
Bruno Jourdain, Christian Vincent and Stéphane Garambois for having constructed bed and surface DEMs using ground penetrating radar
for the bed and drone data for the surface. We also thank Léonard Seydoux for insightful discussions on MPF and SVD.

## Appendix A:  Construction of icequake catalogues


There exists a wide range of seismic sources in glaciers as well as detection schemes (see  Podolskiy and Walter, 2016,  for a
review on the methodology). The processing employed for icequake detections and localizations must be adapted to the type of
network (network versus array, number of sensors, sensor spacing and array aperture) and to the type of the events of interest
which involve various waveforms (e.g. Walter, 2009; Röösli et al., 2014; Podolskiy and Walter, 2016). Dispersive Rayleigh
waves are well suited for investigating the glacier subsurface including the basal interface as they are primarily sensitive to the
S-wave velocity structure and can sample depths up to approximately $1/3$ of the wavelength. We then focus our study on one
class of glacier seismic events being surface icequakes generated by ice crevassing. Another advantage of using such events is
their high rate of time occurence ($\sim 10^2 - 10^3$ recorded events per day) and their potentially wide spatial coverage which is
optimal for the application of seismic interferometry techniques (Walter et al., 2015). We here introduce the methods used to
compute the icequake catalogues at Gornergletscher (Sect. A1 and A2) and Glacier d'Argentière (Sect. A3).

### A1   Icequake detection

Seismic waveforms of surface icequakes generally exhibit a first low-amplitude P-wave followed by impulsive Rayleigh waves
(Fig. 1c). Such events can be detected using a template matching on continuous seismograms (Mikesell et al., 2012). This
cross-correlation method exploits the signal coherency with a reference waveform and can be used on single stations or across
a network. Nevertheless, the most common and straigthforward detection approach is to implement an amplitude threshold
trigger.

The most broadly used algorithm in weak-motion seismology and on glaciers for detecting impulsive events (e.g. Walter,
2009; Canassy et al., 2012; Barruol et al., 2013) is the "short-time-average through long-time-average trigger" referred to as
STA/LTA (Allen, 1978). It continuously calculates the average values of the absolute amplitude of a seismic signal in two
consecutive moving-time windows. The short time window (STA) is sensitive to seismic events while the long time window
(LTA) provides information about the temporal amplitude of seismic noise at the recording site. When the ratio of both exceeds
a preset value at a single station or coherently across a network, an event is declared.

As icequakes usually propagate to a few hundred meter distances before getting attenuated, the amount of identified events
varies with the network configuration and the requested number of stations to trigger concurrently. For the Gornergletscher
study, we work with events that have been detected by running a STA/LTA trigger over 5-15 Hz bandpass filtered continuous





seismograms, using STA windows of 0.3 s (i.e. typical icequake duration), LTA windows 10 times longer and a threshold value of 8. To declare an event, we require at least half of the network stations to pass the trigger value.

## A2 Icequake location

The vast majority of icequakes recorded on glaciers are localized near crevasses that extend no deeper than $\sim 30$ m (Walter
et al., 2008; Lindner et al., 2018a). To locate the events at Gornergletscher, we fix the source depth to the surface and invert for the epicenter distance following the automated approach of Roux et al. (2010) and Walter et al. (2015), also similar to Mikesell et al. (2012). This method employs cross-correlations to automatically measure differences in Rayleigh wave arrival times across the network.

To be able to record coherent icequake waveforms with high enough signal-to-noise ratios (SNR) at couples of sensors, the
network aperture should be less than 1 km, or at least consist of several pairs of stations whose separation is shorter than the distance at which surface waves start to be strongly attenuated in the ice.

In the same spectral band that is used for event detection, icequake signals are first windowed around the Rayleigh wave and cross-correlated for each pair of stations to obtain time delays. Time delay measurements are then refined to subsample precision by fitting a quadratic function to the cross-correlation function centered on its discrete maximum. The best Easting
and Northing coordinates of the source are concurrently inverted with the apparent propagation velocity to match the time delay catalogue for pre-selected pairs of stations. We only consider time shift measurements at pairs of stations whose cross-correlation maximum is above 0.8. This allows us to minimize complex source and/or propagation effects on seismic waveforms and then on observed arrival times that could not be fitted with an oversimplified velocity model.

The inversion process is an iterative procedure using a quasi-Newton scheme (Roux et al., 2010, equation 3). Reliability of
icequake locations varies as a function of the events being inside or outside the network. Using a seismic network similar to the one of Gornergletscher used in the present study, Walter et al. (2015) estimated that, in the azimuthal direction the error remains below 2° for average apparent velocities in the range of 1600-1650 m/s.

## A3 Array processing: matched-field processing using beamforming

A seismic network is called an array if the network aperture is shorter than the correlation radius of the signals that is the
maximum distance between stations at which time series are coherent, i.e. typically less than 1 km for glacier sources recorded by on-ice deployments (Podolskiy and Walter, 2016). A seismic array differs from a local network mainly by the techniques used for data analysis.

Dense sensor arrays have many advantages as the SNR can be improved by summing the individual recordings of the array stations. Compared to a single or couples of sensors, array processing techniques, such as beamforming, allow for time domain
stacking which constructively sums coherent signals over the sensors and cancels out incoherent random noise, enhancing the signal detection capability.

Continuous data are scanned through matched-field processing (MFP) which involves time domain beamforming (Kuperman and Turek, 1997). Beamforming uses the differential travel times of the plane wave front due to a specific apparent slowness



(inverse of velocity) and back azimuth to individual array stations (Rost and Thomas, 2002). If the single-station recordings
are appropriately shifted in time for a certain back azimuth and velocity, all signals with the matching back azimuth and
slowness will sum constructively. MFP can be processed using a decomposition of seismic signals in frequency components.
To be declared as an event and furthermore, with accurate location, the beampower of aligned seismic waveforms at a given
frequency (i.e. the norm of the cross-spectral density matrix and the array response, see Lindner et al., 2018a, equation 3) must
pass a preset trigger threshold. The final event location can be averaged from the beamforming solutions obtained at several
successive discrete frequencies (assuming that slowness and back azimuth are close to constant in the considered frequency
band).

MFP was successfully used by Corciulo et al. (2012) to localize microseismic sources at the exploration scale using ambient-
noise data. Moreover, recent studies focused on developing an automatic, optimization-based MFP approach that does not
require grid-search to localize thousands of weak seismic events in a complex fault zone (Gradon et al., 2019) and hydraulically
fractured area (Chmiel et al., 2019).

The MFP method of Chmiel et al. (2019) was used on Argentière array to detect and locate about 4000 events each day,
with a beampower threshold averaged over frequencies between 5 and 30 Hz set to 0.5. Locations of icequakes recorded
over the 5 week deployment are presented in Fig. 2a. For computing the cross-correlation functions from icequake waveform
described in Sect. 3, we restrict ourselves to 11100 events (red stars in Fig. 12). Such events have been identified following
Lindner et al. (2018a) with a STA/LTA trigger on 8-16 Hz bandpass filtered continuous seismograms (STA= 0.3 s, LTA= 3.6 s,
trigger threshold= 11 for events detected concurrently at the four corner stations of the array). Locations are the beamforming
solutions using a grid search over Easting and Northing positions in 25 m×25 m steps. All events with beampower lower than
0.5 were discarded.

## Appendix B: Computation of phase velocity dispersion curves

Because dispersive surface waves of different frequencies propagate at different speeds, computation of seismic velocities
generally involves Fourier analysis to decompose the wave into frequencies that compose it. One can distinguish two types of
wave speeds.

The phase velocity $c$ is the speed at which the phase of a wave propagates in space and is related to the angular frequency $\omega$
and the wavenumber $k$ as

$$c(\omega) = \frac{\omega}{k(\omega)} \tag{B1}$$

The angular frequency is related to the time periodicity of the signal of frequency $f$ as $\omega = 2\pi f$. The ground displacement is
also periodic in space over a distance equal to the wavelength $\lambda$ that is used to describe how the wave oscillation repeats in
space via the wavenumber $k = 2\pi/\lambda$.

If the harmonic waves of different frequencies propagate with different phase velocities, the velocity at which a wave group
propagates differs from the phase velocity at which individual harmonic waves travel (Stein and Wysession, 2009). The group





velocity $u$ of a wave is the velocity with which the overall envelope shape of the wave amplitudes propagates through space. If the signal has energy over a wide range of frequencies, $u = \mathrm{d}\omega/\mathrm{d}k$ and the group velocity is related to the phase velocity as

$$u(\omega) = c(\omega) + \frac{\mathrm{d}c}{\mathrm{d}k} \tag{B2}$$

In ambient noise tomography, it is common to measure the group velocity of dispersive Rayleigh waves travelling in the
Earth's crust and upper mantle (e.g. Shapiro et al., 2005; Mordret et al., 2013). Dispersion curves of group velocities are usually computed using the Frequency Time Analysis (FTAN) of the noise cross-correlation time-series (Levshin et al., 1992). FTAN employs a system of narrow-band Gaussian filters, with varying central frequency, that do not introduce phase distortion and give a good resolution in the time-frequency domain. For each filter band the envelope of the inverse Fourier transform of the filtered signal is the energy carried by the central frequency component of the original signal. Since the arrival time is
inversely proportional to group velocity, for a known distance, the maximum energy of the time-frequency diagram is obtained as a function of group velocity with frequency.

In glaciers, due to homogeneous ice, only weakly dispersive surface waves are recorded on on-ice seismometers. It is then difficult to use FTAN to measure Rayleigh wave group velocity dispersion. We choose here to compute the phase velocity dispersion curve for Rayleigh waves using different approaches. Obtaining the group velocity from there is then straightforward
while the reverse is not possible due to unknown additive constants which arise from the integration of the phase velocity over frequency (equation B2).

## B1    Array processing: frequency-wavenumber analysis

Frequency-wavenumber analysis (f-k) was used to compute the velocity dispersion curves at Glacier d'Argentière (Sect. 3.1 and 3.2). The f-k analysis is a standard array processing for computing phase velocities from seismic time-series recorded on
a line of receivers (Capon, 1969). It enables to identify and separate wave types and wave modes and also design appropriate f-k filters to remove any seismic energy in the original signal time-series.

The most basic f-k processing employs a 2D Fourier transform on both time and spatial components to construct the f-k diagram (Fig. B1b). We then need to select the dispersion curve of the phase of interest by picking the energy maxima of the 2D Fourier transform output. The absolute value of the f-k space is then transformed into the velocity $c(f)$ via the equation B1
(Fig. B1c).

There exist multiple array techniques for computing frequency-velocity diagrams using spectral analysis in time and space domains. Some of them are described in Rost and Thomas (2002), Gouédard et al. (2008a) and referenced in Ohrnberger et al. (2004). Concerning the cross-correlation functions obtained at Argentière array, we employ the phase-shift method of Park et al. (1998) which allows to construct a frequency-velocity diagram where dispersion trends are identified from the pattern of
energy accumulation in this space. Then, necessary dispersion curves are extracted by following the diagram amplitude trends (Fig. 3c and Fig. B2b). All types of seismic waves propagating horizontally are imaged if they take any significant energy noticeable from the relative intensity of the diagram.





The performance of an array for deriving phase velocity values in a wavenumber or frequency range depends on its geometry and on the wavefield characteristics (i.e. frequency range and magnitude of seismic energy with respect to attenuation). The capability for resolving phase velocity at a given frequency depends on the array aperture (described by the array diameter $\Delta_{\mathrm{max}}$) and minimum sensor spacing ($\Delta_{\mathrm{min}}$) so that at least two wavelengths are sampled between adjacent receivers to avoid aliasing in the wavenumber domain (e.g. Wathelet et al., 2008). Phase velocities should then be computed for frequencies which satisfy $\Delta_{\mathrm{min}} \leq n\lambda \leq \Delta_{\mathrm{max}}$, with $n$ usually taken as 2 or 3. This relationship depends on the expected phase velocity as $\lambda = c/f$.

## B2 Aki's spectral method

This method was used to compute the Rayleigh wave velocity at the GIS and Gornergletscher arrays (Sect. 4.2 and 5.2). Whereas the f-k techniques are based on the assumption of a plane wave arriving at the array, the Aki's spectral method (also referred to SPAC method, Tsai and Moschetti, 2010) bases its theoretical foundation on the precondition of a scalar wavefield which is stationary in both space and time. As detailed below, this technique does not require specific array geometries to compute phase velocities and can be used on single pairs of stations. Another advantage concerns the capability to resolve discrete frequencies on a potential wider range than what is possible using f-k methods as the Aki method produces robust and unbiased measurements at distances smaller than two wavelengths (Ekström et al., 2009). The major limit is set by the seismic wavefield characteristics.

Aki (1957) states that the azimuthally averaged normalized cross-spectrum $S(\Delta, \omega_0)$ for a receiver separation $\Delta$ and frequency $\omega_0$ varies as $J_0$, the zero-order Bessel function of the first kind

$$S(\Delta, \omega_0) = J_0 \left( \frac{\omega_0}{c(\omega_0)} \Delta \right) \tag{B3}$$

This relation suggests that the dispersion curve of phase velocities can be obtained from the fit of a $J_0$ Bessel function to the cross-spectrum obtained on a loop of receivers of same radius $\Delta$, or also, as demonstrated by Cox (1973), to the cross-spectrum obtained for a single station pair if computed on an azimuthally isotropic wavefield.

Ekström et al. (2009) successfully obtain phase velocity estimates at discrete frequencies from ambient noise cross-correlations, by associating the zero-crossing of the real part of the data cross-spectrum with zeros of a Bessel function following equation B3. Preiswerk and Walter (2018) and Lindner et al. (2018b) both use this method to obtain dispersion curves of Rayleigh wave speeds from cross-correlations of on-ice seismic records.

Application of this method is presented in Fig. B3 for one cross-correlation function obtained at Argentière. Because of possible noise contained in the correlation time-serie, the cross-spectrum is first smoothed to avoid any extra zero-crossing measurement. As there is a possibility of having missed one or several zero-crossings (indicated by black squares in b), several dispersion curves are generated (black dashed lines in c). The correct one still needs to be identified by judging the plausibility of the results given the expected velocities of the propagation medium (Ekström et al., 2009).





The dispersion curve estimation can be refined by fitting the entire cross-spectrum with a Bessel function, instead of fitting the zero-crossings only. We develop an approach similar to Menke and Jin (2015) who employ a grid search to generate an initial estimate of the phase velocity that matches the observed cross-spectrum and then use a generalized least-squares procedure to refine this initial estimate.

The prior dispersion curve has to be as close as possible to the measured dispersion curve to avoid cycle skipping during
the fitting. In our procedure, we take as a starting model, the average dispersion curve obtained from f-k analysis of the cross-correlation section computed at Argentière array. For sparse network configuration (as in Greenland and Gornergletscher, i.e. Fig. 9d and 11c), we take the theoretical dispersion curve computed with Geopsy software and based on the prior knowledge we have on the subsurface. The dispersion curves are then modelled as polynomial functions to enforce their smoothness and to reduce the number of fitting parameters and help for the convergence of the fitting process. The order of the polynomial can
be varied. Usually a polynome order of 5 is a good compromise between smoothness and complexity of the dispersion curve. We then use a least-square inversion procedure on the polynome constant coefficients to compute the best dispersion curve which reproduces the observed cross-spectrum.

Fig. B3 shows the output of this procedure which yields to the same dispersion curve (red line in c) as the most probable one computed by fitting the zero-crossings. The overall-fitting method is particularly efficient for estimating more accurate
phase velocities than with the zero-crossing fit, when considering correlation functions with low SNR (Menke and Jin, 2015). However it is particularly sensitive to the frequency range in which the least-square inversion is performed. When considering cross-correlation functions computed in narrow frequency bands as in Sect. 4 and 5, the method introduces strong side effects near the frequency corner limits due to filtering (Fig. 9d and 11c). The cross-spectrum must then be fitted considering carefully selected frequency components.

In the example shown in Fig. B3, the cross-correlation function is computed for frequencies 1-25 Hz. The Bessel fitting method is applied to frequencies above 2 Hz and enables to widen the velocity estimates down to 3 Hz (that is the frequency at which the inter-station distance is approximately equal to one wavelength) when compared to the zero-crossing output.

**B3    Slant-stack technique on discrete sources**

This method employed by Walter et al. (2015) can only be applied on cross-correlation functions that are computed on discrete
sources (i.e. icequakes). It was used to obtain the velocity from ICC at Glacier d'Argentière in section 3.3. We here exploit the phase time-difference in the arrival times of Rayleigh waves with respect to the source position, and reproduce the azimuthal variations of phase times assuming a constant velocity.

The plane wave approximation implies a sinusoidal dependence of the arrival times which depend on the source azimuth and propagation velocity $c$ as $\Delta \cos(\theta)/c$ with $\theta$ here defined as the source azimuth relatively to the station pair axis (Fig. B4
a-b). We call the endfire lobes the two areas aligned with the receiver direction, in which the phase of the correlation function is stationary with respect to azimuth. The angular aperture of the endfire lobes depends on the ratio between the seismic wavelength $\lambda$ and the station separation as $\delta\theta = \sqrt{\lambda/\Delta}$ (Roux et al., 2004; Gouédard et al., 2008b).





To measure seismic velocities at one station path and at different frequencies, we filter the individual correlation functions computed for each event to octave-wide frequency ranges. The lower frequency we can resolve is determined by the icequake

spectral content and is most importantly related to the station separation as we require that at least two wavelengths are sampled. We then restrict the analysis to 15-30 Hz.

We assign all cross-correlations to event azimuth bins of 5° to minimize the effects of location errors. For each trace, the arrival time of the Rayleigh wave is measured as the maximum of the correlation function computed at each frequency. We then invert the best sinusoide fit to the times of the maxima (in the least-square sense). The best solution gives the velocity

estimate at the central frequency of the spectral band.

The velocity solution estimated by this method is naturally averaged over the azimuth range and can only be considered as the average velocity in the presence of strong azimuthal anisotropy which implies azimuthal variations of propagation velocities (Sect. 3.3). Nevertheless, the least-square solution fits very well the Rayleigh wave arrival times in the azimuth range of the stationary phase zones (Fig. B4b), and then is considered to represent well the propagation medium between the two stations.

To minimize the effects of location errors or low SNR of the correlation components, we perform a jackknife test on randomly selected events to fit the sinusoide. We require that the maximum of the cross-correlation stacked over bootstrap samples (i.e. selected correlation functions that have been shifted by the inverted arrival times prior stacking) exceeds 0.7. We require that at least 10 azimuth bins including the endfire lobes are considered in the sinusoidal fit. The final velocity at each frequency is then averaged over 200 jackknife tests.





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

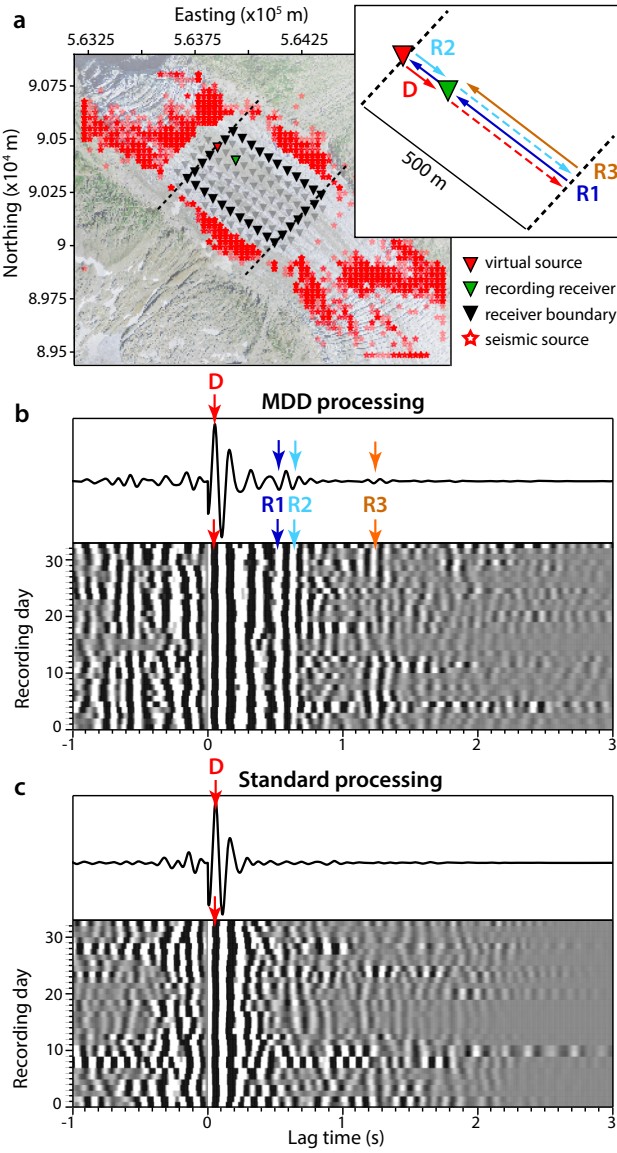

**Figure 12.** (a) Source–receiver geometry used for the computation of icequake virtual-source responses through the application of MDD (b) to the cross-correlations in (c). In this particular case, a receiver contour (consisting of 32 regularly-spaced receivers) and a single receiver between these receiver lines (green triangle) are illuminated by 4282 sources (i.e. icequakes) on either side of the receiver cavity. The receiver colored in red is here turned into a virtual source, whose response is recorded at the green receiver. Due to the reflecting boundary conditions and the receiver geometry, the emitted wave travels back and forth between the two receiver lines indicated with the black dashed lines. We obtain multiple reflections noted $R_i$ ($i$ indicates the number of virtual reflection), which are visible on the MDD correlation gather and averaged stack in (b). Virtual reflections $R_i$ create an artificial coda after the direct Rayleigh wave reconstructed on the GF. This coda is observed to be coherent through time (here we show cross-correlation stacks computed on two-day sliding window with an overlap of one day) and is suitable for CWI studies. (c) is the same as (b) but for standard processing of icequake cross-correlations (ICC).



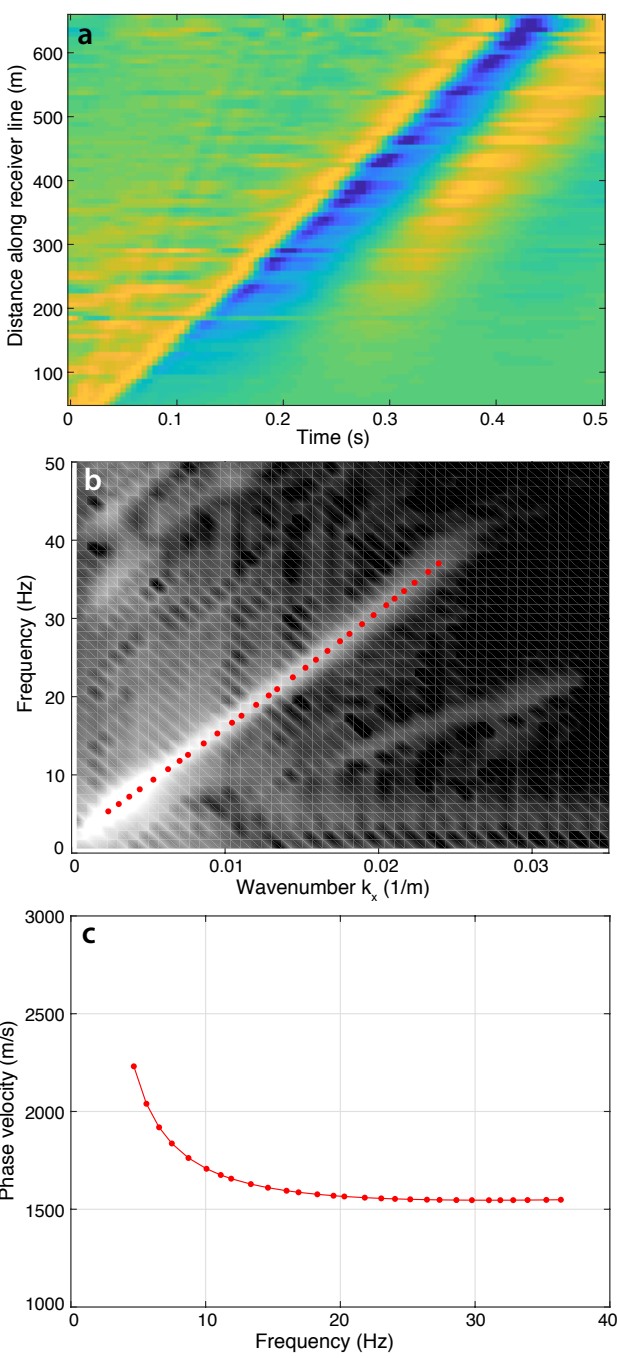

**Figure B1.** Computation of Rayleigh wave phase velocity from f-k analysis of the noise cross-correlation section obtained at Argentière array (a). (b) Frequency-wavenumber diagram obtained from the 2D Fourier transform of (a). Red dots show the peaks of energy maximum at each frequency, and correspond to the Rayleigh wave fundamental mode. (c) Phase velocity dispersion curve obtained from the interpolated peaks in (b).

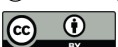
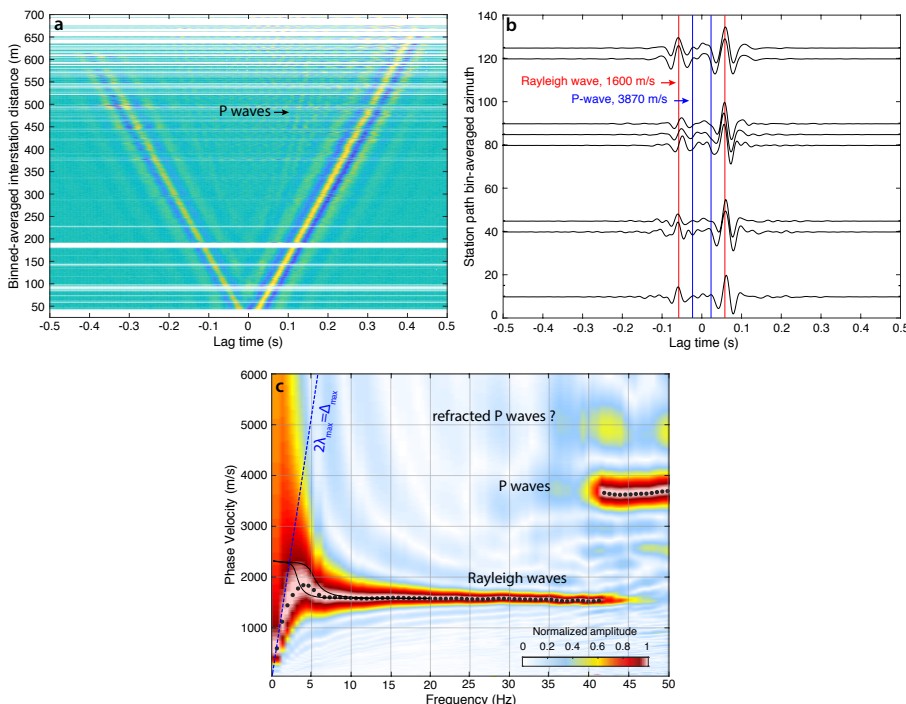

**Figure B2.** Same as Fig. 3 but for icequake-cross-correlations (a) computed in Glacier d'Argentière. (b) Frequency-velocity diagram obtained from f-k analysis of the correlation functions in (a) using the phase-shift method of Park et al. (1998). The extracted dispersion curve of Rayleigh and P-waves are plotted in black dots.





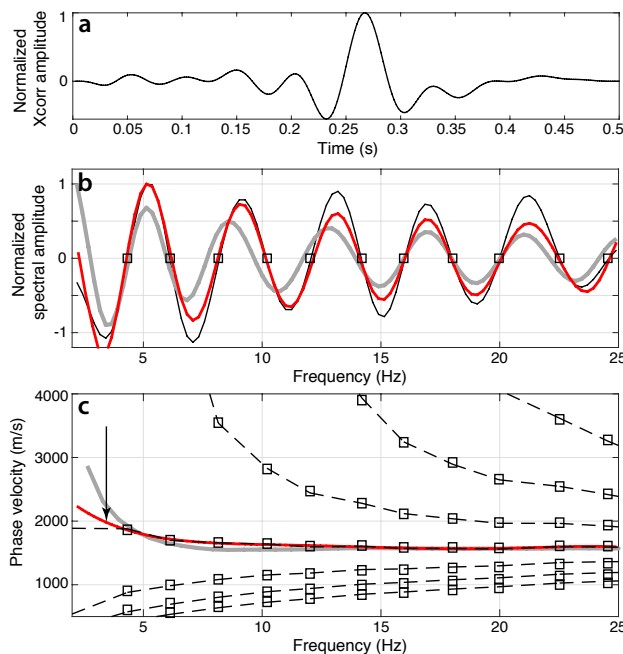

**Figure B3.** Computation of Rayleigh wave phase velocity using Aki's spectral method. (a) Symmetric icequake cross-correlation function obtained for one receiver pair in Glacier d'Argentière, with 450 m apart stations. (b) The real part of the spectrum of (a) is in black and associated zero-crossings are marked by squares. The gray line indicates the real part of the spectrum associated with a priori phase velocity dispersion curve which serves as a starting model for the least-square fit (in red) of the observations. (c) Corresponding phase velocities estimated by zero-crossings (black dashed lines) and least-square fit (red). The prior dispersion curve used for the Bessel fit is plotted in gray. The black arrow indicates the minimum frequency above which we can trust the velocity measurements and corresponds to approximately one wavelength.



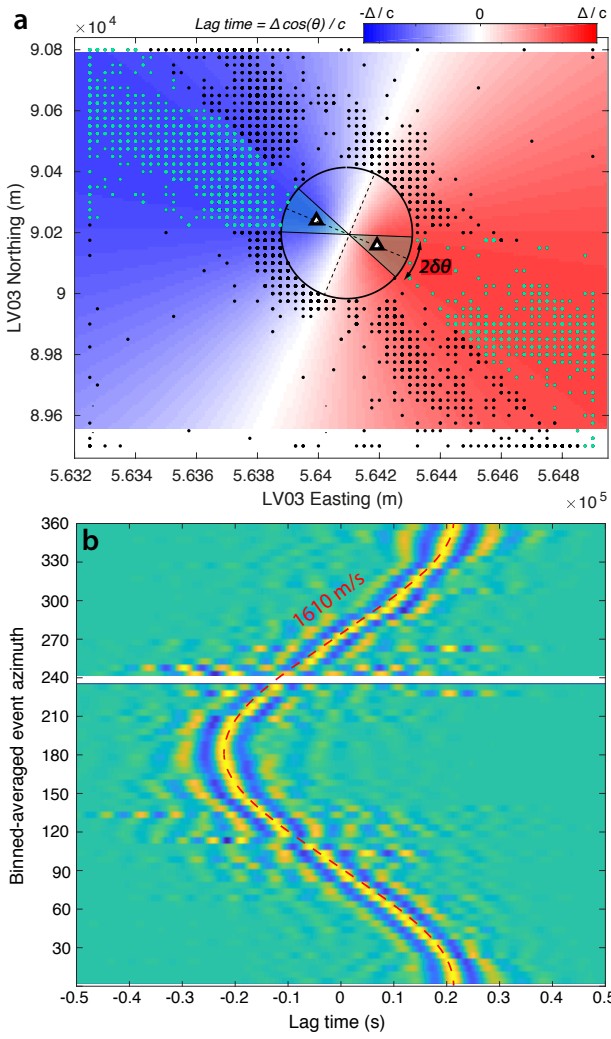

**Figure B4.** (a) Icequake locations (dots) in Glacier d'Argentière whose waveforms are cross-correlated at the two stations (triangles) to obtain (b). Green dots in (a) show the events that lie in the endfire lobes of aperture $2\delta\theta$ (see main text). (b) The icequake cross-correlation functions (here averaged in $5°$ azimuth bins) give coherent arrival times for Rayleigh waves traveling in the ice between the two stations, which are a function of the inter-station distance $\Delta$ and the event azimuth relative to the station path $\theta$, as plotted in background colors in (a). The red dashed line shows the least-square fit of the arrival times with a sinusoidal function of phase velocity $c = 1610$ m/s for central frequency 15 Hz. Correlation functions are here filtered between 10-20 Hz.