# Peer review of "On the Green's function emergence from interferometry of seismic wavefields generated in high-melt glaciers: implications for passive imaging and monitoring"

_The Cryosphere, 2019_

## Referee Comment (RC1) · Naofumi Aso (Referee) · 30 Dec 2019

Sergeant et al. presents multiple approaches to extract Green's functions from the seismic record on the surface of glaciers. Such an effort is challenging and this study looks technically correct, but the manuscript needs several revisions. One potential major problem is the length of the paper. I found many paragraphs are not necessary after spending time to read through the article. Considering that this is not a dissertation nor textbook, reducing unnecessary texts makes the points of this study clearer. In addition to this, I provide specific comments below.

L36–37: The "crustal" and "local scales" do not look contrary. Probably "local to regional" or "regional to global"?

L39: What are the original observations? Please rephrase.

L60: Rather than theoretically, experimentally? If authors want to claim the condition is met THEORETICALLY, please explain which theory.

L64: The statement looks the opposite. Considering that the next sentence is mentioning the source distribution, it sounds better to say: the condition (i) can compensate for the lack of condition (ii).

L80–81: Brief explanation is expected for "virtual reflector" if authors want to mention it here.

L102: waves that => whose seismic waves

L116: Moment magnitude? Local magnitude? Surface-wave magnitude?

Figure 1: Please label all panels: one panel below b and above d.

L123: brief => short

L151: Please explain why the authors use only the vertical component, while the sensor is three-component?

L164: How long was the experiment? It should be explained beforehand.

L166: Please make the sentence complete. GPS stations were deployed, and the DEM is used for later analyses?

L167: What "GPR" stands for?

L170: Insert space before: from

L187–188: because of instrumental sensitivity => because of low instrumental sensitivity

Figure3: ranging from 150 m to 250 m => either 150 or 250 m

L213: anticausal => acausal *Please use the same word consistently throughout the manuscript

L223: Why is it claimed to be correct?

L236: Fig. 3b => 3c?

L240–241: ICC has limited energy at low frequencies just because spectral whitening is not applied?

Figure 5: What is "all" generated misfit values? Isn't it 2500, as explained in the main text? To calculate deviation around each node line, how long horizontal distance did authors consider?

L282: What is "thickness absolute values"? Probably no need to say "absolute values" here.

L284: Why errors and uncertainties are linked to bedrock velocities? Is there any theory ? or is it just implied based on the result? Please explain.

L315: Taking derivatives should not increase the number of cycles of sinusoids in the theory of math. Please rephrase the sentence.

L319: It should be explained earlier that they calculate "within 250 m of the target point".

L323: Please add explan "transversal crevasses" as "perpendicular to flow" here.

L331–332: The sentence of "However we do not exclude ..." is unclear. It sounds like some excuse for something but needs modifications to make it logical, definitely.

L333: aligned-flow => flow-parallel

L350: Short explanation of MFP is needed here. Especially, providing the direct purpose of the processing helps readers. Here, "to locate specific localize sources" would

be appropriate.

L361: "localized" is better than just dominant.

L380: frequency band of => frequency band between

Figure 7: frequency band of => frequency band between

L403: SVD itself is not a process to decrease the number of information, and therefore it is inappropriate to mention "as few coefficients as possible" in this sentence.

L408: Are eigenvectors normalized to be unit?

L409: Please choose either "singular value" or "eigenvalues" to be consistent throughout the manuscript.

L414: threshold in "singular values / eigenvalues"?

L415: Eigenvector is not a scaler, so it is inappropriate to say "eigenvector" above the threshold.

L426–427: Why it is thought to be related to frequency content differences? DIfferent eigenvector does not necessarily mean they consist of different frequency contents.

L427: SVD => eigenvalue?

L427–428: I do not understand why the number of ambient noise sources is more than the number of receivers based on this observation. Please explain this more logically.

L429: reconstructed => decomposed

L430: Please use a different symbol for $K\_i$, which is different from the previous $K\_i$.

L444–446: This explanation is too redundant: i.e., L431–432.

L461: wavenumber vectors are normalized => wavenumber is normalized

L481: Why only one-day used in this study?

L483: Again, it is too straightforward to apply the same analyses to the other day, but why authors are not interested in doing so as a part of this study?

L493: Subtitle should represent what is done in each section. In this sense, the study site "Gornergletscher" is not necessary and instead making it confusing.

L525: Fig 1c => 1d ?

L532: Fig 1c => 1d ?

L540: Do authors see the potential effect of early aftershock in the coda wave?

L542: Please show the coda window to be used in the figure 1d as an example.

L542 Fig 1c => 1d

L546: Does it mean the authors are applying spectral whitening and 1-bit normalization together?

L572–586: Please explain this as a regular paragraph. In addition, many words are redundant (e.g., N or M is already explained).

L613: Indeed, the arrivals could result from the reflection at the bed. Isn't it easy to calculate and validate?

L629: Surprisingly, there are no further analyses of the GF in this section.

L643: allow => allows

L695: Does SH refer to SH wave?

Naofumi Aso @Tokyo Institute of Technology
* * *

---

## Referee Comment (RC2) · Anonymous Referee #2 · 3 Jan 2020

Sergeant et al. present an overview of different methods used to estimate structural information about glaciers and an ice sheet using seismic waves. The main focus of the study is on estimating the direct Rayleigh wave between seismic station using passive recordings of ambient noise and (near-surface) icequakes. The authors present multiple methods (traditional correlation, MFP, MDD, etc.) and highlight the benefits and limitations of each method. The overall goal is to show the usefulness of high-density seismic arrays deployed on ice to image the structure directly beneath the ice, as well as monitoring changes in this structure through time.

[Figure]

The authors demonstrate each of the proposed methods, albeit using dataset from different glaciers. All of the results are quite convincing, but I do question some of the interpretations about why certain features exist in the recovered wavefields. I have listed those below in the comments. In general the paper is well written and clear. In some aspects the text needs to be tightened up though. Statement are not entirely complete or not entirely accurately described. The figure fonts can all be enlarged as well. In general, the actual results in all methods are very nice and I congratulate the authors on recovering such nice GFs from icequake and glacier noise data. This is not easy.

I have listed major comments/concerns below and provide an annotated PDF with many more minor comments.

General:

Paragraph around line 50: Interferometry recovers an approximation to the Green's function/impulse response. There are many assumptions that influence the accuracy of this approximation. This should be better explained or at least noted. Also, the description of the causal and acausal parts of the GF estimate should not be limited to only the direct wave (as is currently done). The more accurate way to characterize what is happening is to describe virtual sources. The direct wave is commonly observed because not all of the assumptions in SI are valid in most field data studies. This paragraph as written is too simplistic and not an accurate depiction of the theory. Please revise to be more complete.

Paragraph starting at line 55: a diffuse or equipartitioned wavefield is not the same thing. Equipartition means that all modes are excited (P,S,Rayleigh,Love,etc.). Diffuse means waves propagating in all directions. This distinction is commonly neglected by most people that write about SI. The current description in this manuscript again confuses these two distinct properties of the wavefield. Please revise throughout the manuscript.

Line 60: The statement that strong scattering exists in the crust is not true at the frequencies commonly used for ambient noise tomography. This statement is very much untrue. At high frequencies (>0.5 Hz) this statement is true, but ambient noise imaging often works because oceans generate the microseismic wavefield all around earth so it is more accurate that condition (i) is met. See the following reference and the papers that have since referenced this paper. Mulargia, F. (2012). The seismic noise wavefield is not diffuse. The Journal of the Acoustical Society of America, 131(4), 2853. https://doi.org/10.1121/1.3689551

The sentence beginning on line 61 is also not entirely accurate. Most studies on glaciers have been unable to reconstruct GFs not because of the lack of scattering but because of the dominant frequency of the background noise. Seismic arrays on glaciers are tiny compared to regional or continental arrays. In order to recover a usable GF in the microseism band you need stations that are more than 1 wavelength apart (neglecting methods like SPAC). When we correlate signals on glaciers in the microseism band the resulting correlations look like autocorrelations because the sensors are pretty much in the same location at the wavelengths of the microseism band. It is more appropriate to state that the noise field lacks the high frequencies needed to generate GFs that contain useful information at the scale of the glacier. If you wish to use icequakes with frequencies above 0.5 Hz, then yes, your statement is accurate, but you should explicitly state this. Everything depends on the frequencies considered and you are neglecting this point in the way that you are writing these statements.

Line 75: You say on line 73 that they do obtain accurate GFs, but then on line 76 you say they don't obtain accurate GFs. Which is it?

Figure 1: fonts are way too small. I also cannot tell which color is "this event" or the "1000 event average". The colors look identical to me. I am assuming the smoother line is the average.

Line 143: Figure 1b –> Figure 1c

Line 166: What is the reason for the partial statement about the 20m resolution DEM? It does not make sense in this sentence. Please read out loud to yourself to see the mistake.

Figure 2: Fonts on axes are again very small.

Appendix A: Line 783: What does "network vs. array" mean?

Line 793: Do you really mean to reference Fig. 1c here? This is a figure of the spectrogram of a GIS signal.

Line 194-195: plane wave approximation –> stationary phase approximation. I do not understand why plane wave is used here. The proper interpretation of the sinusoidal shape is the stationary-phase. See Snieder, R., Van Wijk, K., Haney, M., & Calvert, R. (2008). Cancellation of spurious arrivals in Green's function extraction and the generalized optical theorem. Physical Review E - Statistical, Nonlinear, and Soft Matter Physics, 78(3), 1–8. https://doi.org/10.1103/PhysRevE.78.036606

Line 194: Why are you referencing Fig. B4 before Fig B1, B2, or B3? Please fix the referencing so that things are referenced in order of appearance. It makes reading easier.

Figure 3 caption: Can you please explain why you think the GF converge better in the along-flow direction based on Fig. 3b? I wonder if you are seeing anisotropy in the ice velocities, rather than some sort of convergence related to the strongest noise sources. That reasoning is somewhat counter to your argument for sign-bitting the data. It can be that the density, not the amplitude, of sources is larger in the along flow direction. That would explain differences in convergence, but what you are stated here is not quite correct. Please revise. Also, note that in line 215 you state that the sources are located homogeneously around the array, which implies the density of sources is even with azimuth. Is this true? Did you do beamforming to look at the azimuthal amplitude of incident waves on the array?

Figure 3: Text is again very hard to read.

Figure 3: The dashed blue line is not the array response. That is the frequency-dependent resolution limit. You actually correctly state this in line 238.

Line 226: Don't you mean 1b, not 1a?

Line 227: See annotated PDF. This first sentence can be stated more accurately because not all phases are dispersive. Instead, you are using the f-k domain to identify phases. We just happen to that particular transform a lot for surface wave dispersion, but as you show in your dispersion image, the P wave is not dispersive.

Line 236: Fig. 3b –> Fig. 3c (You should really pay attention to not mislabeling your figures in the future.)

Line 240: Fig. B2b –> Fig. B2c.

Figure 4: axes fonts could be larger

Figure 5: What are the units on the misfit values? Are the misfits the same in (a) and (b)?

Line 258: Fig. 3b –> 3c

Table 1 states that Vs in the granite can be as low as 1000 m/s, but there are not gray lines in Figure 5b that show you tested this velocity. Can you please explain why? I think it would be easy to change the range in Table 1 and not influence the results of the inversion. It actually appears that the lower layer velocity never goes below the upper layer velocity. Is there something in Geopsy that imposes increasing depth and prevents low velocity layers?

Line 271: I do not follow the statement that the ice thickness is 7 to 15 meters thick on the edges. Figure 5c shows ice on the edges more than 100m thick. Can you please explain this discrepancy between the text and the figure? Am I missing something here?

Appendix B: Line 915: SPAC works for single stations when you have an isotropic incident wavefield, otherwise you need an array and averaging. You even state this on line 919 with "azimuthally averaged". You should be careful with your wording in line 915. You are not telling the whole story.

Section 5.1: Do you really need an "origin of coda waves" section? This is already explained with a references in the introduction of the paper. To me this paper is unnecessarily long because everything is explained rather than simply cited.

Line 533: What is your reason to state that the energy is back-scattered? Rayleigh waves have significant forward scattering. See Snieder, R. (1986). 3D linearized scattering of surface waves and a formalism for surface wave holography. Geophysical Journal of the Royal Astronomical . . . , 581–605. Retrieved from http://onlinelibrary.wiley.com/doi/10.1111/j.1365-246X.1986.tb04372.x/abstract, in particular Figures 6 and 7 for example.

Figure 10: Why are the azimuth ranges in (a) and (b) not the same? Are you using difference sources for each station? Or is the azimuth relative to the interstation path, rather than absolute azimuth? I would think the two matrices should be missing the same azimuths if the icequakes used were the same in the two cases. (It is a very nice result by the way!!)

Line 556: Why not beamform the coda? Take the average beam over all time windows. This would highlight illumination problems.

Line 616: What is an anisotropic diffuse wavefield?

Please also note the supplement to this comment:
https://www.the-cryosphere-discuss.net/tc-2019-225/tc-2019-225-RC2-supplement.pdf

**Supplement:**

[revised manuscript text omitted]

---

## Author Comment (AC1) · 4 Feb 2020

Sergeant et al. presents multiple approaches to extract Green's functions from the seismic record on the surface of glaciers. Such an effort is challenging and this study looks technically correct, but the manuscript needs several revisions. One potential major problem is the length of the paper. I found many paragraphs are not necessary after spending time to read through the article. Considering that this is not a dissertation nor textbook, reducing unnecessary texts makes the points of this study clearer. In addition to this, I provide specific comments below.

[Figure]

Thank you for your careful reading and comments. We improved the discussions you were referring to in the comments below. We shortened some parts, revised our writing and used fewer technical terms to smoothen the reading.

**Introduction:**

L36–37: The "crustal" and "local scales" do not look contrary. Probably "local to regional" or "regional to global"?

We modified to "from regional to local". Line 36

L39: What are the original observations? Please rephrase.

We now give additional details on the original observations: "Ambient noise studies have so far led to original observations such as thermal variations of the subsoil, spatio-temporal evolution of the water content, stress changes along fault zones with applications to geomechanics, hydrology and natural hazard". Lines 39-40

L60: Rather than theoretically, experimentally? If authors want to claim the condition is met THEORETICALLY, please explain which theory.

We rephrased the entire paragraph (Lines 65-68) being more precise on theoretical conditions (i.e. equipartitioned source wavefield) and the limitations of their applicability to the Earth which then ends up with the simplified assumption of a diffuse wavefield in practice. We replaced the specific statement you are referring to by: "The latter condition (ii) is sufficiently met only at high- frequency (> 0.5 Hz) in the inhomogeneous Earth's crust. Even if the noise wavefield is not generally diffuse (Mulargia, 2012), the presence of scatterers in the Earth's crust and the generation of oceanic ambient noise all around Earth make ambient noise interferometry applications generally successful."

L64: The statement looks the opposite. Considering that the next sentence is mentioning the source distribution, it sounds better to say: the condition (i) can compensate for the lack of condition (ii).

Thank you for reporting this. Indeed, the statement was incorrect and has been modified.

L80–81: Brief explanation is expected for "virtual reflector" if authors want to mention it here.

Instead of speaking of "virtual reflector seismology", we specifically refer to the process that was used in the referred study as "multidimensional deconvolution on a contour of receivers" and refer to section 6.2. for additional explanation.

**Section 2: Material and data**

L102: waves that => whose seismic waves

We modified as suggested.

L116: Moment magnitude? Local magnitude? Surface-wave magnitude?

We specified local magnitude.

Figure 1: Please label all panels: one panel below b and above d.

We changed the label accordingly.

L123: brief => short

We modified as suggested.

L151: Please explain why the authors use only the vertical component, while the sensor is three-component?

We add a paragraph to explain that using horizontal components from on-ice recordings require additional processing step to reorient the sensors (Lines 158-162).

L164: How long was the experiment? It should be explained beforehand.

The experiment was 5 week-long as stated in Line 164.

L166: Please make the sentence complete. GPS stations were deployed, and the DEM is used for later analyses?

Thank you for reporting this. We rephrased the sentence. Line 174-175

L167: What "GPR" stands for?

We modified GPR to Ground-Penetrating Radar.

L170: Insert space before: from

Space added

**Section 3: Glacier d'Argentière dense array**

L187–188: because of instrumental sensitivity => because of low instrumental sensitivity

We modified as suggested.

Figure3: ranging from 150 m to 250 m => either 150 or 250 m

We modified as suggested.

L213: anticausal => acausal. Please use the same word consistently throughout the Manuscript

Modified to "acausal". We checked for the whole manuscript.

L223: Why is it claimed to be correct?

We want to highlight here the differences in ICC and NCC arising from different source contributions. When compared to NCC, ICC quality is very much less sensitive to the orientation of the station pair because we control the icequake source distribution. We then obtain "correct" GF estimates for ICC, in the sense that spurious arrivals vanish when controlling the icequake source aperture. On the contrary, NCC yield to poor Green's function estimates at some station paths which are not aligned with the most abundant noise sources down/upstream of the array. We rephrased the sentence as: "The control of the icequake source aperture enables to minimize the spurious arrivals which are observed on some NCC (Fig. 3b) and obtain more accurate Rayleigh wave traveltimes at most station paths (Fig. B2b) " Lines 230-231.

L236: Fig. 3b => 3c?

We modified to Fig 3c

L240–241: ICC has limited energy at low frequencies just because spectral whitening is not applied?

ICC and NCC are both computed on spectrally whitened seismograms (for same

frequency corners). The difference in ICC and NCC spectral energy mainly result from the spectral content of icequake seismograms. Indeed, icequakes have short duration of about 0.2 s and then do not carry much energy below 5 Hz. We specify this: "Reconstruction of Rayleigh waves and resolution of their phase velocities using f-k processing are differently sensitive for NCC and ICC at frequencies below 5 Hz (Fig. 3c versus Fig. B2c) as ICC have limited energy at low frequency (Fig. 4a) due to the short and impulsive nature of icequake seismograms (Fig. 1d-e)." Lines 245-247

Figure 5: What is "all" generated misfit values? Isn't it 2500, as explained in the main text? To calculate deviation around each node line, how long horizontal distance did authors consider?

Yes, it is right and we modified the text in the caption accordingly to the main text. Ice thickness uncertainty comes from inversion results reached for misfit values that are below one standard deviation of the 2500 best misfit values. This consideration generally gives a misfit value threshold of 0.02. We added in the main text additional details on how is calculated the misfit (normalized RMS error), Lines 266-267. A maximum misfit value of 0.02 corresponds to ground models which reproduce the data dispersion curve with an approximate error of 2%. The normalizing term for the misfit function is the standard deviation of the uncertainties of the data dispersion curve measured at each frequency. These uncertainties are larger at lower frequencies below 5 Hz because we have less redundant measurements at these wavelengths considering 450m-long profiles. We specify this in Line 273-274.

L282: What is "thickness absolute values"? Probably no need to say "absolute values" here.

"absolute" was removed.

L284: Why errors and uncertainties are linked to bedrock velocities? Is there any theory ? or is it just implied based on the result? Please explain.

Errors and then uncertainties in ice thickness (as derived here from the deviation of best-fitting ground models) are linked to bedrock velocities as stated few lines before (Lines 270-272: "Walter et al. (2015) explored the sensitivity of the basal layer depth to the other model parameters and report a trade-off leading to an increase in inverted ice thickness when increasing both ice and bedrock velocities."). Indeed, using the Greenland ice-sheet array, Walter at al (2015) find a best match for the ice-thickness of 540 m by fitting the Rayleigh wave dispersion curve obtained from match-field processing. They test the sensitivity of the result to the values of S-wave velocities when fixing them in the ice and in the bedrock. However, in our case, the velocities in the ice are very well constrained as all misfit values generally lower than 5% yield to Vs around 1700m/s. We did not perform any further tests like Walter et al, 2015 to assess the ice thickness dependency on the bedrock velocity solely. But from our inversions, we see that the ice thickness estimate slightly depends on the rock S-velocity (Fig 5b). We say on Lines 272-273: "Here the ice thickness estimation is most influenced by the rock velocities as we notice that a 100 m/s increase of basal S-velocity results in an increase in ice thickness up to 15 m" So our statement purely comes from observations and refers to the analysis of Walter et al, 2015. We add additional details Lines 273-374 about the errors and constraints on the bedrock velocities that also related to 3D effects and larger uncertainties on the dispersion curves at lower frequenciesÂă(see also answer above): "These results are moreover influenced by larger errors at lower frequencies (Fig. 5a) which comes from less redundant measurements at great distances."

Walter, F., Roux, P., Röösli, C., Lecointre, A., Kilb, D., and Roux, P.: Using glacier seismicity for phase velocity measurements and Green's function retrieval, Geophysical Journal International, 201, 1722–1737, 2015.

L315: Taking derivatives should not increase the number of cycles of sinusoids in the

theory of math. Please rephrase the sentence.

This is a misleading from the terms we used. We modified "derivation" to "the formulation of equation 1 also gives rise to (...)"

L319: It should be explained earlier that they calculate "within 250 m of the target point".

We moved this sentence earlier to line 327.

L323: Please add explain "transversal crevasses" as "perpendicular to flow" here.

We now specify "transversal crevasses, i.e. perpendicular to the ice flow"

L331–332: The sentence of "However we do not exclude ..." is unclear. It sounds like some excuse for something but needs modifications to make it logical, definitely.

We rephrased the entire paragraph. What we want to point out is that our anisotropy measurements are not punctual but are averaged over the ice column given the consistency of the align-flow fast-axis pattern with frequency. At frequencies 10-15 Hz, the sensitivity kernels are not zero near the base of the glacier (Fig 4b). We then attribute the along-flow fast axis pattern to features at depth, likely a water conduit. We rewrote the paragraph as: "Alignment of the fast-axis directions with that of ice flow appears along the central lines of the glacier (receiver lines 4-5) with anisotropy degrees of 0.5% to 1.5%. This feature is only observed along the deepest part of the glacier where it flows over a basal depression. Results are here computed for seismic measurements at 25 Hz and maps of anisotropy do not change significantly with frequency over the 15-30 Hz range. If we extend our analysis down to 7 Hz, we notice that the aligned-flow fast-axis pattern starts to become visible at 10 Hz. At frequencies lower than 10 Hz, the fast-axis generally tend to align perpendicular

to the glacier flow because lateral topographic gradients introduce 3D effects and non-physical anisotropy. The results presented here are not punctual measurements but are rather averaged over the entire ice column. The vertical sensitivity kernels for Rayleigh waves (Fig. 4b) are not zero in the basal ice layers at the considered frequencies. The align-flow anisotropic pattern is likely attributed to a thin water-filled conduit as also suggested by locations of seismic hydraulic tremors at the study site (Nanni et al, 2019)." Lines 335-340

L333: aligned-flow => flow-parallel

We did not modify the wording.

**Section 4: MFP at the Greenland ice-sheet**

L350: Short explanation of MFP is needed here. Especially, providing the direct purpose of the processing helps readers. Here, "to locate specific localize sources" would be appropriate.

We added the following short explanation: "One of the approaches we apply here is Matched-Field Processing (MFP) (Kuperman and Turek, 1997), which is an array processing technique allowing to locate low-amplitude sources. MFP is similar to a traditional beamforming that is based on phase-delay measurements" Lines 356

L361: "localized" is better than just dominant.

We removed "dominant" from the title of the subsection: L̈ocation of noise sources at the GIS via matched-field processing"

L380: frequency band of => frequency band between

Modified

Figure 7: frequency band of => frequency band between

Modified

L403: SVD itself is not a process to decrease the number of information, and therefore it is inappropriate to mention "as few coefficients as possible" in this sentence.

We corrected the sentence to: "SVD is a decomposition of the CSDM that projects the maximum signal energy into independent coefficients (. . .)."

L408: Are eigenvectors normalized to be unit?

U and V are unitary matrices, so their columns (eigenvectors) are unit vectors (e.g., |eigenvector| = 1) forming an orthonormal basis. We replace "orthogonal matrices" with "unitary matrices".

L409: Please choose either "singular value" or "eigenvalues" to be consistent throughout the manuscript.

We choose "eigenvalues" and modified the text accordingly.

L414: threshold in "singular values / eigenvalues"?

As above.

L415: Eigenvector is not a scalar, so it is inappropriate to say "eigenvector" above the threshold.

We changed the phrase to "the index of eigenvectors".

L426–427: Why it is thought to be related to frequency content differences? Different eigenvector does not necessarily mean they consist of different frequency contents.

We think that the dominant source (the moulin) is located either in the first eigenvector or the second eigenvector depending on the frequency. This can be related to the seismic signature of the tremor and the distinctive frequency bands of either elevated or suppressed seismic energy that can be also noted on the spectrogram of the hydraulic tremor (Figure 1c). We changed the following sentence: "This might be related to the change in the distribution of the dominant sources depending on the frequency related to the seismic signature of the hydraulic tremor and the distinctive frequency bands generated by the moulin activity (Figure 1c)."

L427: SVD => eigenvalue?

We mean the "eigenvalue distribution". We corrected the phrase to: "Moreover, the eigenvalue distribution decays steadily (. . .)."

L427–428: I do not understand why the number of ambient noise sources is more than the number of receivers based on this observation. Please explain this more logically.

We do not claim that the number of noise sources is higher than the number of receivers, but that the number of degrees of freedom (number of parameters that can be used to describe the seismic wavefield, ax explained earlier in the manuscript L:400-405) is higher than the number of receivers, as defined in Seydoux, 2017. This confirms that the wavefield is undersampled by the seismic array (see Seydoux et al, 2017 for details). We changed the phrase into the following: "The latter confirms that the wavefield is undersampled by the seimic array (see Seydoux et al., 2017) for

details)."

L429: reconstructed => decomposed

Here we mean "reconstructed", as we talked about the reconstruction of the CSDM by using individual eigenvectors.

L430: Please use a different symbol for $K_i$, which is different from the previous $K_i$.

Thank you for this comment, we now use the symbol for the reconstructed CSDM.

L444–446: This explanation is too redundant: i.e., L431–432.

We removed the last two repeating phrases.

L461: wavenumber vectors are normalized => wavenumber is normalized We change the phrase to the following: "The wavenumbers $k_x$ and $k_y$ are normalized by the wavenumber $k_0$ corresponding to Rayleigh wave slowness s=1/1680 s/m."

L481: Why only one-day used in this study?

Please see our reply below.

L483: Again, it is too straightforward to apply the same analyses to the other day, but why authors are not interested in doing so as a part of this study?

We agree that it is straightforward to apply the same analysis to another day. For example, Figure R1 shows the same analysis for the 28th of July (the following day after the day presented in our study). Moreover, we can take it even further, and

stack the Green's function estimates for these two days (Figure R2). However, one can see that the moulin seismic signature is located in the second eigenvector for the 27th of July and the first eigenvector for the 28th of July. Therefore, in order to extend this analysis to other days, one should find an automatic criterion to find the index of eigenvectors that corresponds to the moulin. Selecting the eigenvectors manually in order to extend the study to more days would be a tedious task and it would not change the interpretation of the current results. We added the following paragraph in the manuscript, although we do not think that adding the Figures R1, and R2 would enhance the manuscript scientific impact. However, if the Reviewer thinks that this is important, we can add the Figures R1, and R2 in the Appendix. "For example, a similar procedure could be performed on other days and the eigenormalized NCF could be stack over a few days to increase the SNR. However, we verified that the index of eigenvectors corresponding to the moulin changes over days (the moulin can be located in the first, second, third etc. eigenvector). This is the reason why it would be useful to find an automatic criterion of the eigenvalue selection based on the MFP output. However, this is beyond of the scope of this paper."

Figure R1: (a) Location of the dominant noise sources using MFP in the frequency band of 2.5 Hz and 6 Hz (the MFP output is averaged over 30 discrete frequencies). (b), (c) Reconstruction of the CSDM for the 28th of July by using first (in (b)) and second (in (c)) eigenvectors that are related to different noise sources. Each figure represents the MFP gridsearch output calculate for the corresponding eigenvectors. (e) Stacked sections of NCC in the frequency band 2.5-6 Hz. The red line shows the propagation of the Rayleigh waves with velocity of 1680 m/s (also in f). (f) Stacked sections of NCC reconstructed in the frequency band 2.5-6 Hz from the CSDM eigenspectrum equalization.

Figure R2: (a) Stacked sections of NCC in the frequency band 2.5-6 Hz averaged over

two days (the 27th and the 28th of July 2016). The red line shows the propagation of the Rayleigh waves with velocity of 1680 m/s (also in f). (b) Stacked sections of NCC reconstructed in the frequency band from 2.5 to 6 Hz from the CSDM eigenspectrum equalization averaged over two days (the 27th and the 28th of July 2016).

**Section 4: CWI at Gornergletscher**

L493: Subtitle should represent what is done in each section. In this sense, the study site "Gornergletscher" is not necessary and instead making it confusing.

We removed "using the Gornergletscher array" from the title. To be consistent all through the manuscript, we also removed "using the GIS array" from the title of section 4. We now refer to the study sites in the following first section (i.e. "5.1. Icequake coda waves at Gornergletscher").

L525: Fig 1c => 1d ?

We modified to Fig 1d.

L532: Fig 1c => 1d ?

We modified to Fig 1d.

L540: Do authors see the potential effect of early aftershock in the coda wave?

Yes, we must carefully select event codas to avoid anisotropic fields generated by incident waves from aftershocks. The event selection is a first important step as stated in the manuscript. The window-optimization scheme further helps to reduce their potential effects. Aftershock a-effects and cause of spurious arrivals are explained in Lines 598-599 "(...) spurious arrivals at times 0 or later could result from

seismic reflections on the glacier bed beneath the stations, cross-correlations of early aftershocks or other noise sources." and lines 617-620 "the abundance of seismic sources in glaciers often pollutes coda wave seismograms. We often find the situation where ballistic body and surface waves generated by early aftershocks from repetitive and subsequent events (or bed reflections) arrive at the seismic sensor only a few milliseconds after the onset of the first event of interest and therefore fall in its coda window. This typically introduce anisotropic wavefields."

L542: Please show the coda window to be used in the figure 1d as an example.

We indicated the coda window by the gray horizontal bar in Fig 1d.

L542: Fig 1c => 1d

We modified to Fig 1d.

L546: Does it mean the authors are applying spectral whitening and 1-bit normalization together?

The waveforms are first spectrally whitened as stated in Line 534 and then one-bit normalized.

L572–586: Please explain this as a regular paragraph. In addition, many words are redundant (e.g., N or M is already explained).

We rephrase this as a regular paragraph and removed the redundant information.

L613: Indeed, the arrivals could result from the reflection at the bed. Isn't it easy to calculate and validate?

It is not easy to calculate as at the study site, we are very close to the glacier margins. We are sensitive to 3D effects from the lateral margins and also strong basal topographic gradients with about 200 m of depth difference around the array (Walter et al. 2009). Such spurious arrivals may result from the correlation cross-terms from very different reflection arrivals in the coda.

Walter, F., Clinton, J., Deichmann, N., Dreger, D. S., Minson, S., and Funk, M.: Moment tensor inversions of icequakes on Gornergletscher, Switzerland, Bulletin of the Seismological Society of America, 99, 852–870, 2009.

L629: Surprisingly, there are no further analyses of the GF in this section.

We improved the discussion and re-organized it. We already discussed the obtained GF in terms of (1) symmetry, (2) spurious arrivals, (3) (non-)diffuse character of coda waves which arise from single scattering rather than multiply-scattering (medium effect) and non-homogeneously distributed sources and (4) our ability to compute reliable Rayleigh wave dispersion curves. We expose the advantage of CWI to obtain Green's function at seismic arrays where uneven icequake or noise source distributions prevent the Green's function estimation.

**Section 5: Discussion**

L643: allow => allows

Modified

L695: Does SH refer to SH wave?

Yes. Specified

Please also note the supplement to this comment:
https://www.the-cryosphere-discuss.net/tc-2019-225/tc-2019-225-AC1-
supplement.pdf

[Figure]

Figure R1: (a) Location of the dominant noise sources using MFP in the frequency band of 2.5 Hz and 6 Hz (the MFP output is averaged over 30 discrete frequencies). (b), (c) Reconstruction of the CSDM for the 28th of July by using first (in (b)) and second (in (c)) eigenvectors that are related to different noise sources. Each figure represents the MFP gridsearch output calculate for the corresponding eigenvectors. (e) Stacked sections of NCC in the frequency band 2.5-6 Hz. The red line shows the propagation of the Rayleigh waves with velocity of 1680 m/s (also in f). (f) Stacked sections of NCC reconstructed in the frequency band 2.5-6 Hz from the CSDM eigenspectrum equalization.

**Fig. 1.**

[Figure]

[Figure]

**Figure R2**: (a) Stacked sections of NCC in the frequency band 2.5-6 Hz averaged over two days (the 27[th] and the 28[th] of July 2016). The red line shows the propagation of the Rayleigh waves with velocity of 1680 m/s (also in f). (b) Stacked sections of NCC reconstructed in the frequency band from 2.5 to 6 Hz from the CSDM eigenspectrum equalization averaged over two days (the 27[th] and the 28[th] of July 2016).

**Fig. 2.**

---

## Author Comment (AC2) · 4 Feb 2020

Sergeant et al. present an overview of different methods used to estimate structural information about glaciers and an ice sheet using seismic waves. The main focus of the study is on estimating the direct Rayleigh wave between seismic station using passive recordings of ambient noise and (near-surface) icequakes. The authors present multiple methods (traditional correlation, MFP, MDD, etc.) and highlight the benefits and limitations of each method. The overall goal is to show the usefulness of high-density seismic arrays deployed on ice to image the structure directly beneath

the ice, as well as monitoring changes in this structure through time.

The authors demonstrate each of the proposed methods, albeit using dataset from different glaciers. All of the results are quite convincing, but I do question some of the interpretations about why certain features exist in the recovered wavefields. I have listed those below in the comments. In general, the paper is well written and clear. In some aspects the text needs to be tightened up though. Statement are not entirely complete or not entirely accurately described. The figure fonts can all be enlarged as well. In general, the actual results in all methods are very nice and I congratulate the authors on recovering such nice GFs from icequake and glacier noise data. This is not easy.

AS: Thank you for your comment and appreciation. We improved the discussions you were referring to in the comments below (i.e. distributed noise sources in Argentière, anisotropic wavefields from coda waves at Gornergletscher). We shortened some parts, revised our writing and used fewer technical terms to smoothen the reading. We revised the general statements on seismic interferometry with more accurate arguments. We enlarge the fonts on every figure.

**Introduction:**

Paragraph around line 50: Interferometry recovers an approximation to the Green's function/impulse response. There are many assumptions that influence the accuracy of this approximation. This should be better explained or at least noted. Also, the description of the causal and acausal parts of the GF estimate should not be limited to only the direct wave (as is currently done). The more accurate way to characterize what is happening is to describe virtual sources. The direct wave is commonly observed because not all of the assumptions in SI are valid in most field data studies.

This paragraph as written is too simplistic and not an accurate depiction of the theory. Please revise to be more complete.

AS: We rephrased the paragraph with more accurate statements (Lines 48-55). We use the terms "approximate" rather than "reconstruct" the Green's function. We always say "under specific conditions" or "simplified approximations" that refer to the next paragraph which further details the theoretical requirement for an equipartitioned wavefield (which is often not met in reality) and is replaced in practice by the diffusive condition of the seismic wavefield (see comment below). For the assumptions in seismic interferometry, we also refer later to the study of Fichtner et al (2017) which give an accurate overview. We removed the term "direct wave".

Paragraph starting at line 55: a diffuse or equipartitioned wavefield is not the same thing. Equipartition means that all modes are excited (P,S,Rayleigh,Love,etc.). Diffuse means waves propagating in all directions. This distinction is commonly neglected by most people that write about SI. The current description in this manuscript again confuses these two distinct properties of the wavefield. Please revise throughout the manuscript.

AS: We corrected this. We give a correct definition of equipartion. We rewrote the paragraph to say that equipartion is a theoretical requirement for Green's function estimate from interstation correlation, but is not met in practice in the Earth. We then work in simplified approximations of a diffuse wavefield. We modify the paragraph as (Lines 57-63): "In theory, the GF estimate is obtained in media capable of hosting an equipartitioned wavefield, that is random and uncorrelated modes of seismic propagation (P, S, Rayleigh, Love, etc) with same amount of energy. In practice, the equipartition argument has limited applicability to the Earth because non-homogeneously distributed sources, in the forms of ambient noise sources, earthquakes and/or scatterers, prevent the ambient wavefield from being equipartitioned across the entire seismic scale (Fichtner et al, 2017, and references therein).

[Figure]

The GF estimation from interstation correlation therefore usually relies on simplified approximations of diffusive wavefields which can be reached in (i) the presence of equally-distributed sources around the recording network (Wapenaar, 2004; Gouédard et al., 2008b) and/or (ii) in strong-scattering settings as scatterers act like secondary seismic sources and likely create a diffuse homogenized wavefield in all propagation directions (e.g. Hennino et al., 2001; Malcolm et al., 2004; Larose et al., 2008)."

Line 60: The statement that strong scattering exists in the crust is not true at the frequencies commonly used for ambient noise tomography. This statement is very much untrue. At high frequencies (>0.5 Hz) this statement is true, but ambient noise imaging often works because oceans generate the microseismic wavefield all around earth so it is more accurate that condition (i) is met. See the following reference and the papers that have since referenced this paper. Mulargia, F. (2012). The seismic noise wavefield is not diffuse. The Journal of the Acoustical Society of America, 131(4), 2853. https://doi.org/10.1121/1.3689551

AS: Thank you for bringing this up. We modified the paragraph and include your suggestions as (Lines 65-68): "The latter condition (ii) is sufficiently met only at high-frequency (> 0.5 Hz) in the inhomogeneous Earth's crust. Even if the noise wavefield is not generally diffuse (Mulargia, 2012), the presence of scatterers in the Earth's crust and the generation of oceanic ambient noise all around Earth make ambient noise interferometry applications generally successful."

The sentence beginning on line 61 is also not entirely accurate. Most studies on glaciers have been unable to reconstruct GFs not because of the lack of scattering but because of the dominant frequency of the background noise. Seismic arrays on glaciers are tiny compared to regional or continental arrays. In order to recover a usable GF in the microseism band you need stations that are more than 1 wavelength apart (neglecting methods like SPAC). When we correlate signals on glaciers in the

microseism band the resulting correlations look like autocorrelations because the sensors are pretty much in the same location at the wavelengths of the microseism band. It is more appropriate to state that the noise field lacks the high frequencies needed to generate GFs that contain useful information at the scale of the glacier. If you wish to use icequakes with frequencies above 0.5 Hz, then yes, your statement is accurate, but you should explicitly state this. Everything depends on the frequencies considered and you are neglecting this point in the way that you are writing these statements.

AS: That is correct. We add two sentences about these details (Lines 69-72): "In glaciers, the oceanic ambient noise field commonly used in crustal studies lacks the high frequencies needed to generate GFs that contain useful information at the scale of the glacier. To target shallower glaciers and their bed, we must work with other sources such as nearby icequakes and flowing water which excite higher-frequency (> 1 Hz) seismic modes (Sect. 2.1). In this context, the lack of seismic scattering (...)"

Line 75: You say on line 73 that they do obtain accurate GFs, but then on line 76 you say they don't obtain accurate GFs. Which is it?

AS: ndeed, Preiswerk and Walter (2018) were able to compute accurate Green's functions on some glacier settings which presented at that time efficient drainage systems. However, due to changes in the drainage system (localized noise sources which sometimes appeared, or initiation of lake drainage), they could not compute accurate GF at every time scale. We add this late info without going into such details: "However, due to localized noise sources in the drainage system that also change positions over time over the course of the melting season, they could not systematically obtain accurate coherent GF when computed on different time ranges, limiting the applications for glacier monitoring."

**Section 2: Material and data**

Figure 1: fonts are way too small. I also cannot tell which color is "this event" or the "1000 event average". The colors look identical to me. I am assuming the smoother line is the average.

AS: We increased the fonts and modified the color of the red line to blue in Fig 1d.

Line 143: Figure 1b –> Figure 1c

AS: We modified to Fig. 1c.

Line 166: What is the reason for the partial statement about the 20m resolution DEM? It does not make sense in this sentence. Please read out loud to yourself to see the mistake.

AS: We rephrased the sentence which had indeed no meaning. The 20 m resolution of the DEM comes from the spatial interpolation of the GPR tracks.

Figure 2: Fonts on axes are again very small.

AS: We increased the fonts.

Appendix A: Line 783: What does "network vs. array" mean?

AS: We distinguish an array and a network mainly from the processing involved and the size of the array with respect to the considered seismic wavelength. We explain this in a first paragraph in appendix A3. We removed this statement here as it is explained a few lines later.

Line 793: Do you really mean to reference Fig. 1c here? This is a figure of the spectrogram of a GIS signal.

AS: We modified to Fig.1a.

**Section 3: Glacier d'Argentière dense array**

Line 194-195: plane wave approximation –> stationary phase approximation. I do not understand why plane wave is used here. The proper interpretation of the sinusoidal shape is the stationary-phase. See Snieder, R., Van Wijk, K., Haney, M., Calvert, R. (2008). Cancellation of spurious arrivals in Green's function extraction and the generalized optical theorem. Physical Review E - Statistical, Nonlinear, and Soft Matter Physics, 78(3), 1–8. https://doi.org/10.1103/PhysRevE.78.036606

AS: We refer to the plane wave approximation as we restrict ourselves to far-field events. The Green function is recovered when the incident plane waves arrive with equal strength from all azimuths (and with randomly distributed and statistically independent amplitudes, i.e. with homogeneous source distributions). You need same azimuthal angle for same cross-correlation arrivals at both stations for individual sources and this is induced by the plane wave approximation (Gouedard et al, 2008). In contrast, to get accurate GF, you only need stationary phase sources. We did not modify our statement, although we revised the paragraph for better clarity.

Line 194: Why are you referencing Fig. B4 before Fig B1, B2, or B3? Please fix the referencing so that things are referenced in order of appearance. It makes reading easier.

AS: Figure B4 is here to illustrate two things: (1) the source-azimuthal dependency of the reconstructed arrival times in the cross-correlation functions when computed on plane waves (as explained in the main text, Lines 200-210), and (2) the method introduced to measure Rayleigh wave phase velocity from the sinusoidal fit to the azimuthal-dependent arrival times (as described in appendix B3). Figure B4 is then an integral part of the appendices and is more cited here as a supplementary figure

for readers who are not familiar with this. Figure B4 is difficult to be renamed without completely rearranging the appendix which would not completely make sense as the appendix is ordered in the same way as the referencing to the different processing techniques that are used in the main text. We did not modify the referencing to figure B4.

Figure 3 caption: Can you please explain why you think the GF converge better in the along-flow direction based on Fig. 3b? I wonder if you are seeing anisotropy in the ice velocities, rather than some sort of convergence related to the strongest noise sources. That reasoning is somewhat counter to your argument for sign-bitting the data. It can be that the density, not the amplitude, of sources is larger in the along flow direction. That would explain differences in convergence, but what you are stated here is not quite correct. Please revise. Also, note that in line 215 you state that the sources are located homogeneously around the array, which implies the density of sources is even with azimuth. Is this true? Did you do beamforming to look at the azimuthal amplitude of incident waves on the array?

AS: You are right about this. We used misleading terms when speaking about "stronger sources" rather than "more sources". We modified the text and caption accordingly. We modified our statements about distributed noise sources which are, according to our analysis, not distributed homogeneously. We rewrite the paragraph (Lines 219-229) to better explain this. Now in our discussion, we separate spurious arrivals at time 0 at station pairs perpendicular to the flow line, with spurious arrivals at other pairs consti-tuted by faster waves which maybe arise from non-aligned sources and/or anisotropy as you point out. We modified the text as: "More sources downstream are likely gen-erated by faster water flow running into subglacial conduits toward the glacier ice fall (Gimbert et al, 2016, Nanni et al., 2019b). Indeed, looking closer at NCC for individ-ual receiver pairs, we sometimes observe spurious arrivals around time 0 (marked as green dots in Fig. 3b), mostly at stations pairs which are oriented perpendicular to

the glacier flow (i.e. azimuth $0^o \leq \phi \leq 50^o$), indicating that dominant noise sources are located along the flow line. At other station pairs (i.e. azimuth $\psi \sim 90^o$), the reconstructed arrival times are slightly faster than expected. This could be an effect of non-distributed noise sources, or anisotropy introduced by englacial features (Sect. 3.3). This analysis shows that even if the noise sources are not equally distributed in space, averaging the NCC in regular distance intervals on a dense array deployment helps the GF convergence."

Furthermore, we located the noise sources by taking advantage of the causal/acausal amplitude asymmetry and the spurious arrivals (approach similar to Stehly et al, 2009 and Retailleau et al 2017). We find that dominating sources are near the glacier ice fall (downstream) and upstream of the array as stated in the main text, but also along the glacier flow line at the center of the array. This observation also satisfies our expectation for an englacial water conduit along the flow line given the along-flow anisotropic patterns we measure at the array center (Fig.6b and modified text in section 3.3). Nanni et al (AGU Fall Meeting abstract 2019) also looked at the locations of seismic hydraulic tremors with beamforming and their observations are well correlated with our noise source locations. Noise source locations are beyond the topic of the paper as our take-home message is on the advantages of using dense arrays which allow to stack and average the GF estimates (see modified text above). We do not further discuss the noise source locations as it will be part of a future study.

Stehly et al, 2009Ăă: A study of the seismic noise from its long-rangecorrelation properties, JGR.

Retailleau et al, 2017Ăă: Locating microseism sources using spurious arrivals in intercontinental noise correlations, JGR

Nanni et al, 2019Ăă: Mapping the Subglacial Drainage System from Dense Array Seismology: a Multi-method Approach, AGU Fall Meeting abstract.

Figure 3: Text is again very hard to read.

AS: We increased the fonts.

Figure 3: The dashed blue line is not the array response. That is the frequency-dependent resolution limit. You actually correctly state this in line 238.

AS: We modified the caption accordingly to the main text.

Line 226: Don't you mean 1b, not 1a?

AS: We modified to Fig.1b

Line 227: See annotated PDF. This first sentence can be stated more accurately because not all phases are dispersive. Instead, you are using the f-k domain to identify phases. We just happen to that particular transform a lot for surface wave dispersion, but as you show in your dispersion image, the P wave is not dispersive.

AS: Thank you for reporting this. We modified our statement accordinglyÂă(Line 235): "Seismic phases and their velocities can be identified on the frequency-velocity diagram (Fig. 3c, black dots) that is obtained from frequency-wavenumber (f-k) analysis (...)".

Line 236: Fig. 3b –> Fig. 3c (You should really pay attention to not mislabeling your figures in the future.)

AS: We modified to Fig.3c.

Line 240: Fig. B2b –> Fig. B2c.

AS: We modified to Fig B2c.

Figure 4: axes fonts could be larger AS: We increased the fonts.

Figure 5: What are the units on the misfit values? Are the misfits the same in (a) and (b)?

AS: The misfit values are the same in (a) and (b) and are calculated as the normalized RMS of the residuals between dispersion curves. More precisely, the misfit can be defined in Geopsy as the RMS error normalized by the standard deviation provided by the error estimate on the dispersion curve to be fitted. When the maximum misfit value is reached (i.e. 0.1), the dispersion curve is reproduced with an approximate error of 10%. We explain this more accurately in Lines 266-270: "Misfit values correspond here to the root-mean square error on the dispersion curve residuals, normalized by the uncertainty average we obtained from the seismic data extraction (error bars in Fig. 5a). The inversion well resolves the S-wave velocity in the ice layer as all best matching models yield to Vs = 1707 m/s for misfit values below 0.05 meaning that the data dispersion curve is adjusted with an approximate error below 5%."

Line 258: Fig. 3b –> 3c

AS: We modified to Fig 3c.

Table 1 states that Vs in the granite can be as low as 1000 m/s, but there are not gray lines in Figure 5b that show you tested this velocity. Can you please explain why? I think it would be easy to change the range in Table 1 and not influence the results of the inversion. It actually appears that the lower layer velocity never goes below the upper layer velocity. Is there something in Geopsy that imposes increasing depth and

prevents low velocity layers?

AS: This is right, thank you for bringing this up. Indeed, we force Geopsy to explore only increasing velocities with depth, so the lower layer velocities are always higher than the ice velocities. We then have modified the velocity range for the low layer in Table 1, and add this condition to the text, Line 264.

Line 271: I do not follow the statement that the ice thickness is 7 to 15 meters thick on the edges. Figure 5c shows ice on the edges more than 100m thick. Can you please explain this discrepancy between the text and the figure? Am I missing something here?

AS: Indeed, the previous text was maybe unclear. Seismic inversions for the lines on the array edges yield to the determination of three layers: (1) a thin layer of 7-15 meters where we find decreased P-velocities with respect to the ice velocities, (2) the ice layer and (3) the bedrock. We discuss the origin of the low P-velocity zones which could be attributed to snow and more likely to the presence of transversal crevasses which introduce anisotropy that can then be modelled by a slow top layer of a few dozens of meters as shown by Lindner et al, 2008a. We also modified Figure 5c to represent this low velocity layer on the top of the ice. We now better explain this in the main text: "For the receiver lines near the array edges (Lines 1-3 and 8), the inversion yields to a low P-velocity surface layer of thickness 15 m and 7 m respectively, above thicker ice (dashed blue zone in Fig. 5c). (...) This low-velocity surface layer could also at least partially be attributed to the presence of pronounced transversal crevasses (i.e. perpendicular to the receive lines) near the array edges, which do not extend deeper than a few dozens of meters (Van der Veen, 1998) and can be modelled as a slow layer above faster underlying ice (Lindner et al., 2018a)."

Lindner, F., Laske, G., Walter, F., and Doran, A. K.: Crevasse-induced Rayleigh-wave azimuthal anisotropy on Glacier de la Plaine Morte, Switzerland, Annals of Glaciology,

pp. 1–16, 2018a.

Appendix B: Line 915: SPAC works for single stations when you have an isotropic incident wavefield, otherwise you need an array and averaging. You even state this on line 919 with "azimuthally averaged". You should be careful with your wording in line 915. You are not telling the whole story.

AS: Thank you for reporting this. We added the condition of the isotropic wavefield âin Line 915: "this technique does not require specific array geometries to compute phase velocities and can be used on single pairs of stations as long as you are in the presence of an isotropic incident wavefield."

**Section 4: CWI at Gornergletscher**

Section 5.1: Do you really need an "origin of coda waves" section? This is already explained with a references in the introduction of the paper. To me this paper is unnecessarily long because everything is explained rather than simply cited.

AS: We shortened this section and moved some general descriptions with appropriate referencing on the diffusive character of coda waves to the introduction of section 5. We renamed section 5.1 as "Icequake coda waves at Gornergletscher". This section now focuses on the description of icequake coda seismogram.

Line 533: What is your reason to state that the energy is back-scattered? Rayleigh waves have significant forward scattering. See Snieder, R. (1986). 3D linearized scattering of surface waves and a formalism for surface wave holography. Geophysical Journal of the Royal Astronomical, 581–605, in particular Figures 6 and 7 for example.

AS: We did not want to imply anything on the back or forward scattering of the Rayleigh wave propagation mode. We were referring to single versus multiply scattering and

removed the term "back-scattered" accordingly.

Figure 10: Why are the azimuth ranges in (a) and (b) not the same? Are you using difference sources for each station? Or is the azimuth relative to the interstation path, rather than absolute azimuth? I would think the two matrices should be missing the same azimuths if the icequakes used were the same in the two cases. (It is a very nice result by the way!!)

AS: We are essentially using the same sources for the two station paths, except that we exclude events that lie in the vicinity of the stations, i.e. at distances shorter than approximately half of the interstation distance given the considered seismic wavelength (as described in the general section 3). As the first station pair in Fig 10a is closer than the one in Fig 10b, we excluded more events for the computation in Fig 10a. Furthermore, the azimuth is here defined as the event azimuth relative to the station path (and not absolute azimuth). As the two station paths are oriented differently with respect to the source distribution, it results in a different azimuth range. The definition of the azimuth was well stated in the main text. We define it correctly in the figure caption.

Line 556: Why not beamform the coda? Take the average beam over all time windows. This would highlight illumination problems.

AS: We did not try beamforming for the following reasons: (i) The short coda duration and the fast drop of coda coherency across the array prevents for time-lapse beamforming. This is also why CWI fails at some station paths as in here. (ii) Low beamformer resolution given the seismic array and coda coherency. (iii) Given sensitivity kernels of seismic coda, seismic coda are strongly sensitive right beneath the stations, and we do not expect to see any convergence to the illuminated zones given the two points above.

Line 616: What is an anisotropic diffuse wavefield?

AS: With "anisotropic diffuse wavefield" we were referring to the anisotropy of the energy flux which is still observed in the late coda as it is still contains information on the source incidence. To overcome this effect in weak scattering medium, numerical simulations of Paul et al (2005) suggest to work with equally-distributed sources. We modified "anisotropic diffused wavefield" to "the long-lasting anisotropy of the diffuse energy flux". We add these explanationsÂă(Lines 608-610): "Indeed, in weak (or homogeneous) media, the incident energy flux from earthquakes can still dominate the late coda resulting in GF time-asymmetry, provided the sources are located in the same distant region. The CWCC asymmetry is expected to disappear with an isotropic distribution of sources or scatterers around the seismic network.". In general we rearranged and rephrased the whole discussion for more accurate and clearer statements.

Please also note the supplement to this comment: https://www.the-cryosphere-discuss.net/tc-2019-225/tc-2019-225-RC2-supplement.pdf

AS: Thank you for such a careful reading. We took into account all of your suggestions.

Please also note the supplement to this comment:
https://www.the-cryosphere-discuss.net/tc-2019-225/tc-2019-225-AC2-supplement.pdf

---

## Author Response (AR2)

*Revision of manuscript tc-2019-225*

       Dear editor,

I and the co-authors thank you for carefully reading and your appreciation on our manuscript tc-2019-225 entitled "On the Green's function emergence from interferometry of seismic wavefields generated in high-melt glaciers: implications for passive imaging and monitoring". I modified the manuscript accordingly to your comments. As the comments were all minor, I do not provide a list of changes, but the annotated manuscript.

Sincerely,

Amandine Sergeant

**On the Green's function emergence from interferometry of seismic wavefields generated in high-melt glaciers: implications for passive imaging and monitoring**

Amandine Sergeant[1,2], Małgorzata Chmiel[1], Fabian Lindner[1], Fabian Walter[1], Philippe Roux[3], Julien Chaput[4], Florent Gimbert[5], and Aurélien Mordret[3,6]

[1]Laboratory of Hydraulics, Hydrology and Glaciology, ETH Zürich, Zürich, Switzerland
[2]Now at Aix Marseille Univ, CNRS, Centrale Marseille, LMA, France
[3]Université Grenoble Alpes, Université Savoie Mont Blanc, CNRS, IRD, IFSTTAR, ISTerre, 38000 Grenoble, France
[4]Department of Geological Sciences, University of Texas El Paso, El Paso, USA
[5]Université Grenoble Alpes, CNRS, IGE, Grenoble, France
[6]Massachusets Institute of Technology, Boston, USA

**Correspondence:** Amandine Sergeant (sergeant@lma.cnrs-mrs.fr)

**Abstract.** Ambient noise seismology has revolutionized seismic characterization of the Earth's crust from local to global scales. The estimate of the  Green's function (GF) between two receivers, representing the impulse response of elastic media, can be reconstructed via cross-correlation of the ambient noise seismograms. An homogenized wavefield illuminating the propagation medium in all directions is a pre-requesite for obtaining accurate GF. For seismic data recorded on glaciers, this condition imposes strong limitations on GF convergence, because of minimal seismic scattering in homogeneous ice and limitations in network coverage. We address this difficulty by investigating three patterns of seismic wavefields: a favourable distribution of icequakes and noise sources recorded on a dense array of 98 sensors on Glacier d'Argentière (France), a dominant noise source constituted by a moulin within a smaller seismic array on the Greenland  Ice Sheet, and crevasse-generated scattering at Gornergletscher (Switzerland). In Glacier d'Argentière, surface melt routing through englacial channels produces turbulent water flow creating sustained ambient seismic sources and thus favorable conditions for GF estimates. Analysis of the cross-correlation functions reveal non-equally distributed noise sources outside and within the recording network. The dense sampling of sensors allows for spatial averaging and accurate GF estimates when stacked on lines of receivers. The averaged GFs contain high-frequency ($> 30$ Hz) direct and refracted P-waves besides the fundamental mode of dispersive Rayleigh waves above 1 Hz. From velocity measurements , we invert bed properties and depth profiles, and map seismic anisotropy, which is likely introduced by crevassing. In Greenland, we employ an advanced pre-processing scheme which includes match-field processing and eigenspectral equalization of the cross-spectra to remove the moulin source signature and reduce the effect of inhomogeneous wavefields on the GF. At Gornergletscher, cross-correlations of icequake coda waves show evidence for homogenized incident directions of the scattered wavefield. Optimization of coda correlation windows via a bayesian inversion based on the GF cross-coherency and symmetry, 
[revised manuscript text omitted]